# Branching Diffusion for Point Processes in Time and Space

**Chao Yang** [1]  **Wenjie Shen** [1]  **Shuang Li** [1]

## Abstract

We propose a non-autoregressive branching diffusion model for generating spatio-temporal point processes. Starting from a geometric principle—the Wasserstein-Fisher-Rao (WFR) gradient flow of a generalized KL divergence toward a simple reference intensity—we obtain a tractable forward noising mechanism with two interpretable components: (i) a Langevin-type *drift-diffusion* step that perturbs event locations and times, and (ii) a *birth-death branching* step that changes the event count via location-dependent thinning (deaths) and Poisson offspring replication (births). We learn the reverse-time dynamics using a permutation-equivariant denoiser that predicts a drift field and a net-growth field, and we train it using an entropic-regularized unbalanced optimal transport (UOT), which naturally handles count mismatch between noisy and clean samples. The resulting generator produces complete spatio-temporal event sets without autoregressive simulation or explicit intensity normalization.

## 1 Introduction

Many event-driven phenomena are naturally modeled as temporal or spatio–temporal point processes (TPPs/STPPs), including social-media activity (Farajtabar et al., 2017), policy interventions over time (Hızlı et al., 2023; Yang et al., 2025b;a), and crime (Dong et al., 2024) or earthquake occurrences (Yuan et al., 2023) in space–time. A flexible *generative* model for such processes is important not only for forecasting, but also for simulation-centric tasks such as exploring counterfactual event patterns, comparing alternative scenarios, and generating realistic synthetic data for downstream analysis.

Most existing point-process models are parameterized

---
[1]School of Data Science, The Chinese University of Hong Kong, Shenzhen. Correspondence to: Shuang Li <lishuang@cuhk.edu.cn>.

*Proceedings of the 43rd International Conference on Machine Learning*, Seoul, South Korea. PMLR 306, 2026. Copyright 2026 by the author(s).

through a (conditional) intensity function and trained by maximizing a likelihood or a surrogate objective. While this intensity-based paradigm underlies strong baselines such as RMTPP (Du et al., 2016), THP (Zuo et al., 2020), Deep-STPP (Zhou et al., 2022), NSTPP (Chen et al., 2020), and SMASH (Li et al., 2023), it has structural limitations from a generative standpoint. First, generation based on conditional intensity function is usually *autoregressive*: events must be simulated one by one via thinning or inversion (Ogata, 1981), causing computational cost to scale with the number of events and tying sampling to a specific intensity parameterization. Second, goodness-of-fit is assessed at the level of local intensities rather than joint event patterns, so long-range temporal dependence, spatial clustering, and event-count statistics are only indirectly controlled. The closest prior work to ours that relax autoregressivity are Add–Thin (Lüdke et al., 2023) and diffusion-based STPP models (Yuan et al., 2023), which typically rely on the forward kernels and loss functions that are not explicitly grounded in the geometry of event patterns or in the mass-variation structure intrinsic to random event counts. In particular, Add-Thin noising reaches a Poisson reference by thinning data events and superposing Poisson noise, which leaves surviving events unchanged while permanently discarding the locations of removed events. This makes it unclear how to incorporate a geometry-aware, continuous corruption of spatio-temporal event locations.

In this work, we take a geometric approach and propose a diffusion framework for spatio-temporal point processes derived from *unbalanced optimal transport*, accommodating both noising of events' time/space locations and event counts in a unified, principled manner. We design the forward process of the intensity function as the Wasserstein-Fisher-Rao (WFR) gradient flow of a generalized KL divergence toward a simple reference intensity. The induced dynamics admit a transparent split into: (i) a Langevin transport operator that continuously redistributes intensity over time and space while preserving total event counts, and (ii) a local source-sink term that creates or annihilates mass, thereby adjusting the expected event count.

We realize this intensity evolution at the sample-path level via a *branching diffusion* on counting measures: atoms follow Langevin drift-diffusion updates and independently undergo birth/death events. Concretely, each forward step

alternates two interpretable operations. A transport step perturbs event locations and times through a Langevin-type update targeting the reference intensity. A subsequent birth-death step adjusts cardinality: each event dies or generates offspring according to locally defined growth rates. In the small-step regime, this stochastic particle system recovers the WFR gradient flow of the underlying variational objective at the level of the mean intensity. Moreover, the forward process converges to the reference process, so generation can start from a simple reference distribution without any autoregressive construction.

The reverse generative model is formulated as a permutation-equivariant (Zaheer et al., 2017) denoiser acting on noisy event sets. At each reverse step, the model predicts, in parallel for all events, both a transport field and a net growth field that approximate the time-reversed drift/diffusion-source/sink dynamics. This yields a fully non-autoregressive sampler that iteratively refines entire event sets rather than generating events sequentially.

For training, we measure the discrepancy between a predicted counting measure and the target counting measure. To accommodate variable event counts, we adopt an entropically regularized unbalanced optimal transport loss (Séjourné et al., 2019), which is differentiable and scalable, and provides a stable denoising objective. This choice mirrors the unbalanced transport structure used to design the forward process; equivalently, it can be viewed as a finite-sample, smoothed analogue of the WFR metric, aligning estimation with the same "transport + mass-variation" principle that motivates the model.

We evaluate the resulting framework on a diverse suite of temporal and spatio–temporal benchmarks. For TPPs, we consider multimodal inhomogeneous Poisson and Hawkes-process synthetics, as well as real Twitter and COVID-19 policy data. For STPPs, we study synthetic processes with evolving spatial modes and two real datasets—Crime and Earthquake—that exhibit complex spatial clustering and temporal dynamics. Generation quality is assessed using Wasserstein distance on event counts, maximum mean discrepancy (Gretton et al., 2012) on event sequences, and a spatio–temporal Sinkhorn divergence that jointly measures temporal, spatial, and count fidelity.

These experiments show that, the proposed method accurately recovers multimodal temporal and spatio-temporal structures, closely matches empirical intensity patterns on real datasets, and achieves competitive or superior performance relative to autoregressive, flow-based, and diffusion-based baselines. Ablation studies highlight the importance of the geometric design: replacing unbalanced transport with balanced variants, removing the debiasing term, or ignoring the spatio–temporal cost structure leads to substantial degradation in sample quality. Moreover, the branching

diffusion remains robust under severe temporal and spatial missingness, extrapolating plausible event patterns from partially observed data.

Overall, this work introduces a geometrically grounded, non-autoregressive diffusion framework for temporal and spatio–temporal point processes. By unifying transport, diffusion, and mass variation within a single statistical construction, the proposed approach provides a principled alternative to intensity-based generative modeling.

## 2 Preliminaries

**Time index convention.** We use $t \in \mathcal{T}$ for the *physical* event-time coordinate of the point process, and $\tau \in [0, 1]$ for the *diffusion/noising* time index of the forward process. We write $s \in \mathcal{S}$ for location and $z = (s, t) \in \mathcal{Z} = \mathcal{S} \times \mathcal{T}$ for a spatio-temporal pair.

### 2.1 Spatio-temporal point processes

Let $\mathcal{S} \subseteq \mathbb{R}^d$ denote the spatial domain and $\mathcal{T} = [0, T]$ a time horizon, and define the spatio-temporal domain $\mathcal{Z} := \mathcal{S} \times \mathcal{T}$ equipped with its Borel $\sigma$-algebra. A *spatio-temporal point process* on $\mathcal{Z}$ can be defined as a random (finite) counting measure $N$ on $\mathcal{Z}$. Equivalently, $N$ is a random element taking values in the space $\mathcal{N}(\mathcal{Z})$ of finite counting measures. For any Borel set $A \subseteq \mathcal{Z}$, the random variable $N(A)$ counts the number of events that fall in $A$. Almost surely, $N$ is purely atomic and admits the representation

$$N = \sum_{i=1}^{N(\mathcal{Z})} \delta_{z_i}, \qquad z_i = (s_i, t_i) \in \mathcal{Z},$$

where $N(\mathcal{Z})$ is the total number of events and $\delta_z$ denotes the Dirac measure at $z$. We write $|N| := N(\mathcal{Z})$ for the event count. Thus, learning a spatio-temporal point process can be viewed as learning a probability law on $\mathcal{N}(\mathcal{Z})$, i.e., a distribution over finite counting measures.

We will also use the associated *intensity* function $\lambda : \mathcal{Z} \to \mathbb{R}_+$, which is a finite non-negative measure on $\mathcal{Z}$ that represents the expected event counts on every measurable sets in the spatio-temporal domain $\mathcal{Z}$.

### 2.2 Diffusion models

Diffusion models (Sohl-Dickstein et al., 2015) couple a *forward* noising process that gradually maps data to a simple reference law with a learned *reverse* process that approximately inverts this corruption. Recalling that $\tau$ denotes the time index in a diffusion process, the forward dynamics is specified by an Itô SDE

$$\mathrm{d}X_\tau = f_\tau(X_\tau)\,\mathrm{d}\tau + g_\tau(X_\tau)\,\mathrm{d}W_\tau,$$

whose time-marginals $\{p_\tau\}$ satisfy the Fokker–Planck equation, which describes conservation of mass under advection and diffusion and reads (with $G_\tau := g_\tau g_\tau^\top$):

$$\partial_\tau p_\tau = -\nabla \cdot (f_\tau p_\tau) + \frac{1}{2} \sum_{i,j} \partial_{x_i} \partial_{x_j} \big((G_\tau)_{ij}\, p_\tau\big).$$

To generate samples, one simulates the *reverse-time* dynamics. Score-based and denoising formulations exploit the fact that the reverse-time drift can be written in terms of the forward drift and the score $\nabla_x \log p_\tau(x)$ (Song et al., 2020), enabling a neural network approximation of the reverse transition. In discrete time, this amounts to parameterizing a reverse kernel $p_\theta(x_{k-1} \mid x_k)$ and fitting it with tractable denoising objectives. A key practical requirement is that the chosen forward transitions are easy to sample and sufficiently tractable to support efficient training. The reverse model is then trained via simple denoising regression: sample a time step $\tau$, draw a noisy state $x_\tau$ from the known forward kernel, and train a network to predict either the clean data $x_0$ or an equivalent target at time $\tau$.

### 2.3 Gradient-flow viewpoint of diffusions on Euclidean space

Many diffusions admit an alternative characterization as *gradient flows* of an energy functional (often a KL divergence) on the space of probability measures equipped with the 2-Wasserstein (optimal transport) geometry. This viewpoint clarifies long-time behavior and provides a principled route to designing forward noising dynamics. For example, consider overdamped Langevin dynamics with target $\nu(x) \propto e^{-V(x)}$ and a time-dependent mobility $\eta(\tau) \geq 0$,

$$\mathrm{d}X_\tau = -\frac{\eta(\tau)}{2} \nabla V(X_\tau)\, \mathrm{d}\tau + \sqrt{\eta(\tau)}\, \mathrm{d}W_\tau,$$
$$\partial_\tau p_\tau = \frac{\eta(\tau)}{2} \nabla \cdot \Big(p_\tau \nabla \log \tfrac{p_\tau}{\nu}\Big),$$

which is precisely the Wasserstein gradient flow of $\mathrm{KL}(p_\tau \| \nu)$, with $\eta(\tau)$ acting as a time rescaling. As a concrete instance, taking $V(x) = \frac{1}{2}\|x\|^2$ yields the variance-preserving forward SDE

$$\mathrm{d}X_\tau = -\tfrac{1}{2}\eta(\tau) X_\tau\, \mathrm{d}\tau + \sqrt{\eta(\tau)}\, \mathrm{d}W_\tau.$$

If $\eta(\tau)$ is held constant on each interval $[\tau_{k-1}, \tau_k]$, the exact Gaussian transition can be written as

$$x_k = \sqrt{\alpha_k}\, x_{k-1} + \sqrt{1 - \alpha_k}\, \varepsilon_k, \quad \varepsilon_k \sim \mathcal{N}(0, I),$$

with $\alpha_k = \exp\big(-\int_{\tau_{k-1}}^{\tau_k} \eta(\tau)\, \mathrm{d}\tau\big)$, which matches the variance-preserving DDPM kernel. (Ho et al., 2020)

## 3 A principled design of the forward process

Euclidean diffusion models evolve a fixed-dimensional state and thus induce a flow of probability laws whose total mass is identically one. In contrast, spatio-temporal point process data are naturally represented as (random) finite counting measures $N$ on $\mathcal{Z} = \mathcal{S} \times \mathcal{T}$, where the event count $|N| = N(\mathcal{Z})$ is random. As a consequence, two samples can contain different numbers of atoms, so there is no fixed ambient dimension. This variable-event-count structure complicates both (i) the design of forward noising transitions that interpolate meaningfully between samples and (ii) the definition of losses that compare noisy and clean samples when their event counts differ.

In this section, we derive a principled forward evolution at the level of the *intensity function*. Concretely, for a counting-measure-valued process $\{N^\tau(z)\}$, we work with a nonnegative function $\lambda^\tau : \mathcal{Z} \to \mathbb{R}_+$ such that, for every measurable set $A \subseteq \mathcal{Z}$,

$$\mathbb{E}\big[N^\tau(A)\big] = \int_A \lambda^\tau(z)\mathrm{d}z.$$

Here, we emphasize that $N^\tau$ is a random counting measure over the spatio-temporal domain $\mathcal{Z}$ at diffusion time $\tau$, and $\lambda^\tau$ is its intensity. To distinguish from physical event time $t \in \mathcal{T}$, we use the superscript $\tau$ to index the time of a diffusion process.

We construct a forward evolution for $\lambda^\tau$ by extending the gradient-flow viewpoint to *finite intensities* using an *unbalanced optimal transport* geometry. (Chizat et al., 2018) Specifically, we combine (i) a generalized KL functional for finite intensities and (ii) the Wasserstein-Fisher-Rao (WFR) metric from unbalanced optimal transport. The resulting WFR gradient flow of generalized KL yields dynamics with an interpretable *Langevin + source-sink* structure: the Langevin component moves mass in $\mathcal{Z}$, while the source-sink component changes total mass, matching the fact that point process samples have random event counts.

### 3.1 Generalized KL divergence for finite intensities

Let $\lambda, \nu : \mathcal{Z} \to \mathbb{R}_+$ be nonnegative integrable functions, where $\nu$ is a fixed *reference intensity* on $\mathcal{Z}$ (e.g., a simple reference intensity to which the forward process is attracted). We use the generalized KL functional

$$\mathrm{KL}(\lambda \| \nu) := \int_{\mathcal{Z}} \Big(\lambda(z) \log \frac{\lambda(z)}{\nu(z)} - \lambda(z) + \nu(z)\Big)\, dz, \quad (1)$$

with the convention that $\mathrm{KL}(\lambda \| \nu) = +\infty$ if $\lambda(z) > 0$ on a set where $\nu(z) = 0$. When $\int \lambda = \int \nu$, (1) reduces (up to an additive constant) to the usual KL between normalized intensities. For general finite intensities, the additional mass terms $-\int \lambda + \int \nu$ preserve nonnegativity and make $\mathrm{KL}(\lambda \| \nu)$ compatible with unbalanced transport formulations in which total mass may vary. In what follows, $\nu$ is fixed and $\lambda^\tau$ is evolved by the forward process.

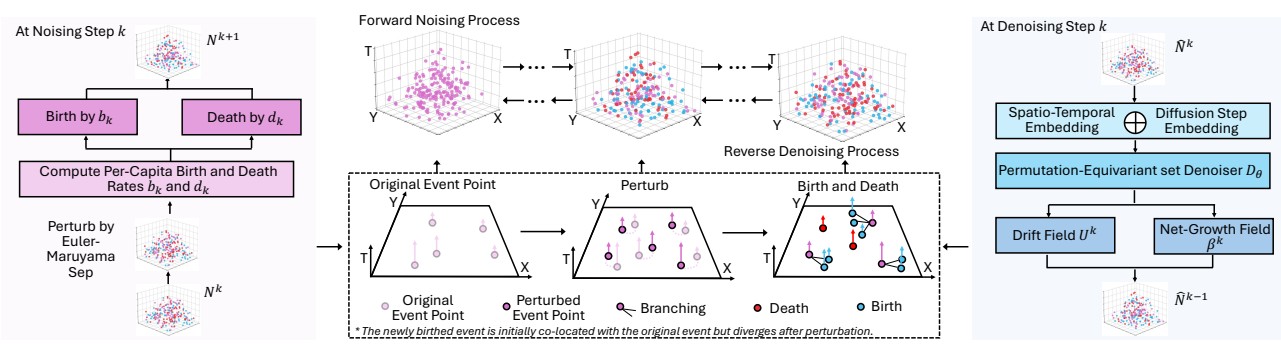

*Figure 1.* Model framework. The purple shaded area denotes the steps of the forward noising process; the blue shaded area shows the steps of the reverse denoising process; and the dotted box outlines the steps of branching diffusion.

## 3.2 Wasserstein-Fisher-Rao (WFR) metric

Balanced optimal transport compares objects of equal total mass and is therefore not well aligned with point processes when the total event count is random and informative. We instead use unbalanced optimal transport, specifically the Wasserstein-Fisher-Rao (WFR) metric, which couples transport with local mass creation and annihilation.

We use the dynamic (Benamou-Brenier-type) formulation. Let $\{\lambda^\tau\}_{\tau \in [0,1]}$ be a diffusion-time-indexed family of non-negative intensities on $\mathcal{Z}$, $v_\tau : \mathcal{Z} \to \mathbb{R}^{d+1}$ a velocity field over $\mathcal{Z}$, and $\alpha_\tau : \mathcal{Z} \to \mathbb{R}$ a growth field. They are coupled by the continuity equation with a multiplicative source term,

$$\partial_\tau \lambda^\tau + \nabla \cdot (\lambda^\tau v_\tau) = \alpha_\tau \lambda^\tau, \quad (2)$$

interpreted in the weak sense (here $\nabla$ and $\nabla\cdot$ act on the coordinates of $z = (s, t) \in \mathcal{Z}$). The WFR metric is defined by

$$\mathrm{WFR}^2(\lambda^0, \lambda^1)$$
$$:= \inf_{\lambda^\tau, v_\tau, \alpha_\tau} \left\{ \int_0^1 \int_{\mathcal{Z}} \left( \|v_\tau(z)\|^2 + \alpha_\tau(z)^2 \right) \lambda^\tau(z) \, dz \, d\tau : \right.$$
$$\left. (2) \text{ holds, } \lambda^{\tau=0} = \lambda^0, \ \lambda^{\tau=1} = \lambda^1 \right\}. \quad (3)$$

The integrand $\|v_\tau(z)\|^2$ is the quadratic transport cost associated with moving mass across $\mathcal{Z}$, while $\alpha_\tau(z)^2$ is the quadratic mass-variation cost associated with local creation and annihilation. Unlike balanced Wasserstein distances, WFR remains meaningful when $\int \lambda^0 \neq \int \lambda^1$.

## 3.3 The WFR gradient flow of generalized KL

We now combine the generalized KL functional with the WFR geometry to obtain a principled continuous-time forward evolution. Specifically, we consider the energy

$$\mathcal{F}(\lambda) := \mathrm{KL}(\lambda \| \nu),$$

and derive its (formal) WFR gradient flow. Intuitively, this yields dynamics that (i) transport mass in $\mathcal{Z}$ toward the reference intensity $\nu$ and (ii) adjust total mass so that $\int \lambda^\tau$ can drift toward $\int \nu$.

The *first variation* of $\mathcal{F}$ at $\lambda$ is

$$\frac{\delta \mathcal{F}}{\delta \lambda}(z) = \log \frac{\lambda(z)}{\nu(z)}. \quad (4)$$

Under the WFR geometry, a tangent direction at $\lambda$ can be represented as

$$\xi = -\nabla \cdot (\lambda v) + \alpha \lambda,$$

with squared norm $\int (\|v\|^2 + \alpha^2) \lambda \, dz$. The WFR steepest descent direction for $\mathcal{F}$ is obtained by selecting

$$v = -\nabla \phi, \quad \alpha = -\phi, \quad \text{where } \phi := \frac{\delta \mathcal{F}}{\delta \lambda}. \quad (5)$$

Substituting (5) into (2) yields the WFR gradient flow equation, which we summarize in the following proposition.

**Proposition 3.1** (WFR gradient flow of generalized KL). *Let $\nu$ be a fixed reference intensity on $\mathcal{Z}$. Formally, the WFR gradient flow of $\mathcal{F}(\lambda) = \mathrm{KL}(\lambda \| \nu)$ satisfies*

$$\partial_\tau \lambda^\tau(z) = \nabla \cdot \left( \lambda^\tau(z) \nabla_z \log \frac{\lambda^\tau(z)}{\nu(z)} \right) - \lambda^\tau(z) \log \frac{\lambda^\tau(z)}{\nu(z)}. \quad (6)$$

*Moreover, $\lambda = \nu$ is the limiting distribution of (6).*

Proposition 6 shows that the WFR gradient flow of generalized KL yields a natural forward evolution for finite intensities that combines transport and mass variation. The reference intensity $\nu$ acts as an attractor, and the dynamics adapt both the spatio-temporal distribution and the total expected event count to approach $\nu$ as $\tau \to 1$.

*Remark* 3.2. When the state space $\mathcal{Z}$ is a bounded domain, the WFR-driven drift-diffusion component of the intensity dynamics is naturally posed with a Neumann-type reflecting/no-flux boundary condition. For the drift-diffusion part of (12), the associated flux is

$$J_\tau := \lambda_\tau u_\tau - \nabla \lambda_\tau, \quad J_\tau(z) \cdot n(z) = 0 \ \text{for } z \in \partial \mathcal{Z},$$

where $n(z)$ is the outward unit normal. At the particle level, this corresponds to a reflected SDE (with an additional boundary local-time term) that keeps trajectories inside $\mathcal{Z}$. In discrete time diffusions, we approximate this reflected dynamics by applying *mirror reflection*. Under standard regularity and ergodicity conditions (e.g., $\mathcal{Z}$ bounded with regular boundary and $\nu$ smooth, strictly positive on $\overline{\mathcal{Z}}$, and integrable), the reflected drift-diffusion with drift $\nabla \log \nu$ is reversible with respect to the measure $\nu(z)\mathbf{1}_{\mathcal{Z}}(z)\mathrm{d}z$. Consequently, its limiting (invariant) distribution is the restriction of $\nu$ to $\mathcal{Z}$.

Expanding the transport term gives the equivalent form

$$\partial_\tau \lambda^\tau(z) = \Delta\lambda^\tau(z) - \nabla \cdot \left(\lambda^\tau(z)\nabla_z \log \nu(z)\right)$$
$$- \lambda^\tau(z) \log \frac{\lambda^\tau(z)}{\nu(z)}. \qquad (7)$$

Define the net growth field

$$\alpha_\tau(z) := -\log \frac{\lambda^\tau(z)}{\nu(z)},$$

so that (7) can be written as

$$\partial_\tau \lambda^\tau(z) = \underbrace{\left(\Delta\lambda^\tau(z) - \nabla \cdot \left(\lambda^\tau(z)\nabla_z \log \nu(z)\right)\right)}_{\text{Langevin}}$$
$$+ \underbrace{\alpha_\tau(z)\,\lambda^\tau(z)}_{\text{source-sink}}. \qquad (8)$$

The Langevin operator is mass-preserving, whereas the source-sink term increases intensity where $\lambda^\tau < \nu$ and decreases it where $\lambda^\tau > \nu$, allowing the expected event count and the spatio-temporal distribution to adapt toward the reference.

### 3.4 Counting-measure realization via branching diffusion

While (8) evolves an intensity $\lambda^\tau$, point process samples are counting measures with unit atoms. A natural sample-path model consistent with the decomposition in (8) is a *branching diffusion* on counting measures: atoms move in $\mathcal{Z}$ according to the Langevin operator, and atoms independently give birth and die with per-capita rates

$$b_\tau(z) := [\alpha_\tau(z)]_+, \quad d_\tau(z) := [-\alpha_\tau(z)]_+,$$

respectively, where $[x]_+ := \max\{x, 0\}$. The induced evolution of the mean intensity takes the form

$$\partial_\tau \lambda^\tau = \mathcal{L}^*\lambda^\tau + (b_\tau - d_\tau)\lambda^\tau, \qquad (9)$$

where $\mathcal{L}^*$ is the adjoint of the Langevin generator. Since $\alpha_\tau(z) = -\log(\lambda^\tau(z)/\nu(z))$ depends on the current intensity, this should be understood as a mean-field interpretation: the deterministic intensity evolution follows (8), while

counting-measure sample paths realize the same transport and mass-variation effects through stochastic motion and birth/death events.

We approximate this evolution in discrete time using an operator splitting, which perturbs atoms via Langevin dynamics and then applies birth-death branching updates. Specifically, let $\{\tau_k\}_{k=0}^K$ be a diffusion-time grid on $[0, 1]$ with increments $\eta_k := \tau_k - \tau_{k-1}$. We define a forward Markov chain $q(N^k \mid N^{k-1})$ on counting measures (with $N_0 \sim p_{\text{data}}$) by applying a Langevin perturbation followed by a birth-death update.

**Perturb step.** Given $N = \sum_{i=1}^M \delta_{z_i}$, define the transport update by a single Euler-Maruyama step of overdamped Langevin targeting $\nu$:

$$z_i \leftarrow z_i + \eta_k\,\nabla \log \nu(z_i) + \sqrt{2\eta_k}\,\varepsilon_i, \quad \varepsilon_i \sim \mathcal{N}(0, I),$$
$$(10)$$

and set $\tilde{N}_k := \sum_{i=1}^M \delta_{\tilde{z}_i}$. If $\mathcal{Z}$ imposes domain constraints (e.g., bounded space $s \in \mathcal{S}$ or bounded event time $t \in \mathcal{H}$), we apply the mirror reflection so that all updated points remain in $\mathcal{Z}$. When $\nu$ is spatio-temporally homogeneous, the drift term vanishes and (10) reduces to Gaussian perturbations.

**Birth-death step.** On the perturbed counting measure $\sum_{i=1}^M \delta_{z_i}$, we apply a birth-death update for duration $\gamma\eta_k$ with net growth field $\alpha_k(z)$ and per-capita birth and death rates

$$b_k(z) := [\alpha_k(z)]_+, \quad d_k(z) := [-\alpha_k(z)]_+. \qquad (11)$$

Each atom at $z_i$ is updated independently as:

- *(Death)* dies with probability $1 - \exp\left(-\gamma\eta_k d_k(z_i)\right)$

- *(Birth)* conditional on survival, sample an offspring count $C_i \sim \text{Poisson}\left(\gamma\eta_k b_k(z_i)\right)$ and add $C_i$ new atoms.

This step discretizes the multiplicative source–sink term $\alpha_t(z)p_t(z)$ in the intensity evolution (9). Interpreting $\alpha_k(z)$ as a local *net* growth rate, the decomposition (11) enforces nonnegative birth and death rates while preserving $b_k(z) - d_k(z) = \alpha_k(z)$ pointwise. Over a short duration $\gamma\eta_k$, an atom at location $z$ experiences death with hazard $d_k(z)$ (yielding survival probability $\exp(-d_k(z)\gamma\eta_k)$) and produces offspring according to a Poisson clock of rate $b_k(z)$ (yielding $\text{Poisson}(b_k(z)\gamma\eta_k)$ births). Since the source-sink term is local in $z$, offspring are created at the parent location, and subsequent transport updates account for spatial spreading.

**Forward-process hyperparameters.** We discretize diffusion time with a grid $\{\tau_k\}_{k=0}^K$ and step sizes $\eta_k = \tau_k - \tau_{k-1}$,

and apply an operator-splitting update: a Langevin move step (10) followed by a birth-death branching step (11). Unless otherwise stated, we use a uniform grid $\eta_k = 1/K$. The scalar $\gamma$ controls the relative strength of the birth–death update compared to the location perturbation; we set $\gamma$ so that per-step birth/death probabilities remain in a stable small-step regime (details in Appendix C).

**Net-growth field and reference intensity.** The ideal net growth field $\alpha_\tau(z) = -\log(\lambda^\tau(z)/\nu(z))$ depends on the unknown intermediate intensity $\lambda^\tau$, so we use a tractable schedule. In our experiments we use a global field $\alpha_k(z) \equiv a_k$ chosen so that the *expected event count* follows a smooth schedule from the initial count toward the reference mass $M_\nu = \int_{\mathcal{Z}} \nu(z)\,dz$ (see Appendix C for the exact form and variants). We choose the reference intensity $\nu$ to be simple (typically spatio-temporally homogeneous on $\mathcal{Z}$ with mass matched to the data), so that the move step reduces to isotropic Gaussian perturbations (with reflection on bounded domains).

## 4 Learning the reverse process

Given the forward Markov chain of counting measures $\{N^k\}_{k=0}^K$ constructed in Section 3, we learn a reverse-time generative model such that ancestral sampling from $N^K \sim \nu$ yields $\hat{N}_\theta^0$ distributed like the data. The overall model framework is presented in Fig. 1. As in diffusion models, we train the model through *denoising*: we generate counting measures $N_k$ by the known forward process, predict a less noisy object, and fit the predictor using a tractable regression objective.

### 4.1 Denoising process

The reverse-time intensity evolution induced by the WFR forward process (6) can be written in another drift-diffusion SDE with birth-death (a standard derivation is deferred to Appendix B):

$$\partial_\tau \lambda_\tau = -\nabla \cdot (\lambda_\tau u_\tau) + \Delta \lambda_\tau + \lambda_\tau r_\tau. \quad (12)$$

with drift

$$u_\tau(z) := 2\nabla \log \lambda_\tau(z) - \nabla \log \nu(z),$$

and net growth field

$$r_\tau(z) := \log \frac{\lambda_\tau(z)}{\nu(z)}.$$

Equation (12) motivates a denoising transition composed of (i) a drift-diffusion step with drift $u_\tau$ and (ii) a birth-death correction driven by the net growth field $r_\tau$. In practice, $\lambda_\tau$ is unknown and we approximate $(u_\tau, r_\tau)$ with a learned denoiser.

### 4.2 Denoiser architecture

Let $N^k = \sum_{i=1}^{|N^k|} \delta_{z_i^k}$ be the counting measure at step $k$ (diffusion time $\tau_k$), with atoms $z_i^k \in \mathcal{Z}$ (e.g., $z = (x,t) \in \mathcal{S} \times \mathcal{T}$). We parameterize the reverse transition via a permutation-equivariant set denoiser $D_\theta$ that maps the current counting measure to (i) a drift/velocity field and (ii) a net-growth field:

$$(U^k, \beta^k) = D_\theta(N^k, k), \quad (13)$$

where

$$U^k = \{u_i^k\}_{i=1}^{|N^k|}, \quad \beta^k = \{\beta_i^k\}_{i=1}^{|N^k|}$$

and we interpret these outputs as pointwise evaluations

$$u_i^k = u_\theta(z_i^k; N^k, \tau_k) \in \mathbb{R}^{\dim(\mathcal{Z})},$$
$$\beta_i^k = \beta_\theta(z_i^k; N^k, \tau_k) \in \mathbb{R}, \quad (14)$$

where $u_\theta$ and $\beta_\theta$ approximate the reverse drift $u_{\tau_k}$ in reverse net-growth $r_{\tau_k}$ in (12), respectively.

In line with set-based denoisers, $D_\theta$ can be implemented as a permutation-equivariant set network over a set of points in $\mathcal{Z}$, using: (i) an embedding of the continuous coordinates $z$, (ii) a diffusion-step embedding for $k$ (or $\tau_k$).

### 4.3 Training loss: entropic unbalanced OT

Let $\mu$ and $\hat{\mu}$ be two (finite) counting measures on $\mathcal{Z}$,

$$\mu = \sum_{i=1}^n a_i\,\delta_{z_i}, \qquad \hat{\mu} = \sum_{j=1}^m b_j\,\delta_{\hat{z}_j}, \qquad a_i, b_j \geq 0,$$

and let $c : \mathcal{Z} \times \mathcal{Z} \to \mathbb{R}_+$ be a ground cost. Denote by $C \in \mathbb{R}_+^{n \times m}$ the cost matrix $C_{ij} = c(z_i, \hat{z}_j)$. The entropic-regularized unbalanced optimal transport (UOT) is defined as

$$\min_{Q \geq 0} \langle Q, C \rangle + \tau \mathrm{KL}(Q\mathbf{1} \| a) + \tau \mathrm{KL}(Q^\top \mathbf{1} \| b) + \varepsilon \mathrm{KL}(Q \| ab^\top)$$
$$(\text{UOT})$$

where $a = (a_i)_{i=1}^n$, $b = (b_j)_{j=1}^m$, and $\mathbf{1}$ is an all-ones vector. The parameter $\tau > 0$ controls the penalty for *mass variation* through the two marginal KL terms. This is the static analogue of the Fisher-Rao contribution in the dynamic WFR formulation introduced in Section 3.2: smaller $\tau$ makes birth-death cheaper. The parameter $\varepsilon > 0$ is introduced for differentiability: it makes the optimal coupling $Q_\varepsilon^\star$ well-behaved for gradient-based learning and enables an scalable Sinkhorn solver; taking $\varepsilon \downarrow 0$ recovers the corresponding unregularized UOT problem.

We compare the predicted measure $\hat{N}_\theta^k$ against the forward-simulated target $N^k$ by solving (UOT) and weight it by step-dependent weights $w_k$. The overall training objective is to find $\theta$ that minimizes

$$L(\theta) = \mathbb{E}_{\substack{N^0 \sim p_{\text{data}}, k \sim \text{Unif}[K] \\ N^k \sim q(\cdot | N^{k-1})}} \left[ w_k \text{UOT}(\hat{N}_\theta^k, N^k) \right]. \quad (15)$$

The training algorithm is summarized in Alg. 3, Appendix A.

## 5 Experiments

Having established a WFR-based diffusion framework on finite spatio-temporal measures and its corresponding reverse-time dynamics, we now empirically evaluate whether this geometrically grounded formulation yields faithful generation of point process configurations. We instantiate the reverse dynamics with a permutation-equivariant set denoiser that acts directly on event sets, enabling joint refinement of event locations and times together with parallel birth–death updates over variable-cardinality configurations. Noisy samples are generated by the forward branching diffusion, and learning proceeds via a denoising objective based on a debiased entropic unbalanced optimal transport (UOT) Sinkhorn divergence. Through a series of experiments on temporal and spatio-temporal point process benchmarks, we aim to assess whether the proposed configuration-level diffusion model can accurately capture event geometry, event-count statistics, and complex dependencies beyond autoregressive intensity-based baselines.

### 5.1 Experimental Setup

The following provides a brief introduction to the datasets, baselines, and evaluation metrics; further details can be found in Appendix D.

**Datasets** For TPPs, we consider two synthetic datasets with known generative structure: **Syn-TPP-1**, a multi-modal inhomogeneous Poisson process, and **Syn-TPP-2**, a Hawkes process with self-excitation; and two real-world datasets: **Twitter**[1] (Farajtabar et al., 2017; Lüdke et al., 2023) and **Covid-19 Policy**[2] dataset (Hale et al., 2021; 2020). For STPPs, we use two synthetic benchmarks with evolving spatial modes: **Syn-STPP-1**, a multi-modal inhomogeneous Poisson process, and **Syn-STPP-2**, a spatio-temporal Hawkes process. Real-world STPP datasets include **Crime**[3] and **Earthquake**[4] dataset.

**Baselines** For TPP generation, we compare against representative likelihood-based and neural models, including **RMTPP** (Du et al., 2016), **THP** (Zuo et al., 2020), and **TriTPP** (Shchur et al., 2020). We additionally include **Add-Thin** (Lüdke et al., 2023), a diffusion-based non-

[1] https://github.com/davecasp/add-thin/tree/main/data
[2] https://github.com/OxCGRT/covid-policy-dataset
[3] http://www.atlantapd.org/
[4] https://earthquake.usgs.gov/earthquakes/search/

autoregressive model, as the closest baseline to our approach. For STPPs, we compare with SOTA neural models designed for spatio-temporal point processes, including **DeepSTPP** (Zhou et al., 2022), **NSTPP** (Chen et al., 2020), **DSTPP** (Yuan et al., 2023), and **SMASH** (Li et al., 2023).

**Metrics** We evaluate generation quality using the **Wasserstein distance (WD)** on event counts and the **Maximum Mean Discrepancy (MMD)** between event sequences, following standard practice in point process generation. In addition, consistent with our configuration-level formulation, we assess joint spatio-temporal fidelity using the **Spatio-Temporal Sinkhorn Divergence (STSD)** between event measures, which jointly captures discrepancies in time, space, and event count.

### 5.2 Results and Analysis

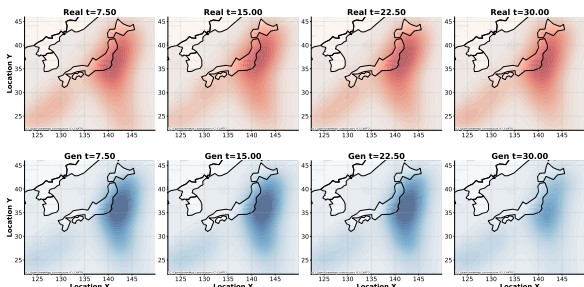

*Figure 2.* Spatio-temporal intensity recovery on the Earthquake dataset.

**Analysis 1: Can the model preserve multimodal spatio-temporal intensity structures and support conditional generation?** We evaluate the model's ability to recover spatio-temporal intensity on both synthetic TPP and STPP datasets. As reported in Fig. 5 and Fig. 6 (Appendix E.1), the proposed method accurately preserves multimodal intensity structures across time and space. On the more challenging real-world Earthquake dataset (Fig. 2), the generated spatial intensities across different time slices are nearly indistinguishable from the empirical ones, faithfully capturing both spatial concentration and temporal evolution. These results demonstrate that the model generalizes beyond synthetic settings and can recover complex real-world spatio-temporal patterns.

In addition, our framework naturally supports conditional generation. Given contextual information $c$, the model generates event sequences from the conditional distribution $p(\mathcal{S} \mid c)$. On synthetic TPP and STPP datasets, the recovered conditional intensities closely match the ground truth (Fig. 7 and Fig. 8, Appendix E.2). On the real-world Earthquake dataset, conditioning on earthquake magnitude leads to clearly differentiated intensity patterns in both spatial location and overall intensity level, indicating that the model

effectively captures context-dependent spatio-temporal dynamics.

**Analysis 2: How does the proposed method compare with SOTA baselines on temporal and spatio-temporal point processes?** As shown in Tab. 1, on TPP benchmarks, our method achieves performance comparable to the strongest existing baselines. Across synthetic datasets, it attains near-optimal MMD and STSD scores, indicating accurate recovery of temporal event distributions.

In spatio-temporal point process settings, the advantages of our approach become more pronounced. Our model consistently achieves the lowest STSD on synthetic STPP benchmarks, indicating superior recovery of evolving spatial modes over time. On real-world datasets such as Crime and Earthquake, it remains competitive with or outperforms strong neural STPP baselines, particularly in terms of joint spatio-temporal fidelity. These results highlight the benefit of our model for capturing global spatio-temporal structure and event-count variability, which are difficult to model using local or sequential generation mechanisms.

**Analysis 3: Which design choices are critical for stable training and faithful generation?** We ablate key model design choices to assess their role in maintaining geometric consistency during diffusion. Shown in Tab. 2, replacing unbalanced optimal transport (UOT) with balanced OT has a limited effect on Syn-TPP-1, but leads to a substantial degradation on Syn-STPP-1, where STSD increases from 0.33 to 0.48, indicating a breakdown in joint spatio-temporal structure. Note that we use the debiased (`UOT`) throughout previous experiments, removing the debiasing terms in the Sinkhorn divergence further degrades performance, with MMD on Syn-TPP-1 increasing from 0.41 to 0.92, reflecting biased and unstable supervision under event-count mismatch. Disabling temporal–spatial cost scaling causes additional performance drops on Syn-STPP-1, highlighting the necessity of properly calibrating time–space geometry. Together, these results show that the proposed components are not optional heuristics, but are essential for preserving the underlying geometry of point process configurations.

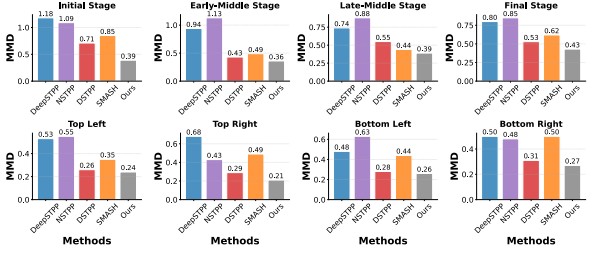

*Figure 3.* Generation performance comparison with different temporal and spatial missing patterns for Syn-STPP-1 dataset.

**Analysis 4: Can the model generate plausible and consistent event sequences under partial observations?** We evaluate the robustness of the proposed method under partial observations on Syn-STPP-1, considering both temporal and spatial missing patterns during training (Fig. 3). Across all settings, our model consistently achieves the lowest MMD, indicating strong robustness to incomplete data.

Under temporal missing patterns, our method remains stable even in the most challenging initial-stage setting where early events are entirely unobserved, achieving an MMD of 0.29 and substantially outperforming DeepSTPP, NSTPP, and DSTPP. Similar advantages are observed across early-middle, late-middle, and final-stage missing regimes. Under spatially missing regions, the performance gap remains pronounced: for example, in the top-right missing region, our model attains an MMD of 0.11 and consistently outperforms all baselines across other spatial patterns.

These results indicate that the proposed configuration-level diffusion model does not rely on specific temporal segments or spatial regions, but instead learns a globally coherent spatio-temporal structure. By operating on unordered event configurations and leveraging unbalanced optimal transport, the model naturally accommodates missing mass and extrapolates plausible event sets, enabling faithful generation under severe temporal and spatial incompleteness.

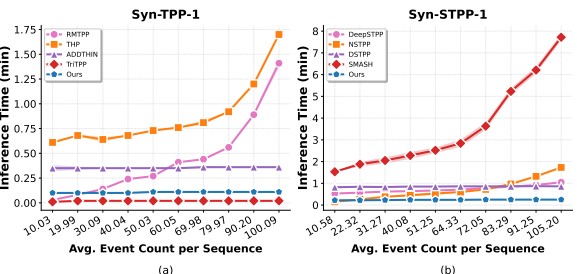

*Figure 4.* Sampling runtime for 200 sequences with varying sequence length of training dataset. All the experiments are conducted over three random runs and the standard error is reflected in the shaded areas. (a) Comparison on Syn-TPP-1. (b) Comparison on Syn-STPP-1.

**Analysis 5: Can the model bring inference time advantages?** TPP/STPP models differ fundamentally in how they generate sequences. Sequential/autoregressive models (e.g., RMTPP, THP, DeepSTPP, and SMASH) generate events one by one, conditioning each event on the previously generated history; accordingly, their sampling cost grows with sequence length. In contrast, full-sequence methods generate the entire sample path jointly, without explicit event-by-event sampling. This category includes diffusion-based models (e.g., Add-Thin and DSTPP), flow-based models (e.g., TriTPP), and our method.

Results in Fig. 4 are consistent with the discussion in Add-

*Table 1.* Comparison of different methods on TPP and STPP datasets. Performance metrics are averaged across three different runs, which reported as (Mean ± SD). The best performance is in bold and also colored in purple.

| Methods | Temporal Point Process Datasets | | | | | | | | | | | |
|---|---|---|---|---|---|---|---|---|---|---|---|---|
| | Syn-TPP-1 | | | Syn-TPP-2 | | | Twitter | | | Covid-19 | | |
| | WD ↓ | MMD ↓ | STSD ↓ | WD ↓ | MMD ↓ | STSD ↓ | WD ↓ | MMD ↓ | STSD ↓ | WD ↓ | MMD ↓ | STSD ↓ |
| RMTPP | $11.85_{\pm0.39}$ | $0.61_{\pm0.01}$ | $12.90_{\pm0.40}$ | $28.62_{\pm0.41}$ | $0.50_{\pm0.00}$ | $29.17_{\pm0.41}$ | $8.86_{\pm0.54}$ | $0.51_{\pm0.01}$ | $10.60_{\pm0.39}$ | $12.73_{\pm0.55}$ | $0.51_{\pm0.00}$ | $14.84_{\pm0.46}$ |
| THP | $5.11_{\pm0.22}$ | $0.57_{\pm0.01}$ | $3.62_{\pm0.33}$ | $16.33_{\pm0.31}$ | $0.67_{\pm0.00}$ | $17.05_{\pm0.32}$ | $7.80_{\pm0.70}$ | $0.43_{\pm0.01}$ | $5.08_{\pm0.56}$ | $16.12_{\pm0.31}$ | $0.41_{\pm0.00}$ | $18.08_{\pm0.30}$ |
| Add-Thin | $1.78_{\pm0.44}$ | $0.05_{\pm0.01}$ | $0.014_{\pm0.000}$ | $\mathbf{2.33_{\pm0.31}}$ | $\mathbf{0.03_{\pm0.00}}$ | $0.03_{\pm0.00}$ | $\mathbf{1.17_{\pm0.13}}$ | $\mathbf{0.05_{\pm0.00}}$ | $2.17_{\pm0.16}$ | $0.80_{\pm0.12}$ | $0.42_{\pm0.02}$ | $\mathbf{0.05_{\pm0.00}}$ |
| TriTPP | $\mathbf{1.07_{\pm0.33}}$ | $0.04_{\pm0.01}$ | $0.013_{\pm0.002}$ | $2.47_{\pm0.14}$ | $0.04_{\pm0.00}$ | $0.02_{\pm0.00}$ | $4.51_{\pm0.69}$ | $0.22_{\pm0.01}$ | $0.32_{\pm0.04}$ | $0.52_{\pm0.32}$ | $0.33_{\pm0.05}$ | $38.33_{\pm9.13}$ |
| Ours* | $2.68_{\pm0.33}$ | $\mathbf{0.04_{\pm0.00}}$ | $\mathbf{0.013_{\pm0.001}}$ | $3.09_{\pm0.29}$ | $0.06_{\pm0.00}$ | $\mathbf{0.01_{\pm0.00}}$ | $6.78_{\pm0.23}$ | $0.11_{\pm0.01}$ | $\mathbf{0.24_{\pm0.02}}$ | $\mathbf{0.37_{\pm0.08}}$ | $\mathbf{0.30_{\pm0.13}}$ | $0.21_{\pm0.18}$ |
| Methods | Spatio-Temporal Point Process Datasets | | | | | | | | | | | |
| | Syn-STPP-1 | | | Syn-STPP-2 | | | Crime | | | Earthquake | | |
| | WD ↓ | MMD ↓ | STSD ↓ | WD ↓ | MMD ↓ | STSD ↓ | WD ↓ | MMD ↓ | STSD ↓ | WD ↓ | MMD ↓ | STSD ↓ |
| DeepSTPP | $8.73_{\pm2.08}$ | $0.41_{\pm0.03}$ | $5.23_{\pm1.18}$ | $18.46_{\pm3.51}$ | $0.91_{\pm0.22}$ | $4.55_{\pm1.25}$ | $31.93_{\pm5.28}$ | $0.40_{\pm0.08}$ | $1.67_{\pm0.48}$ | $15.77_{\pm4.29}$ | $1.33_{\pm0.21}$ | $6.34_{\pm1.74}$ |
| NSTPP | $7.32_{\pm1.35}$ | $0.47_{\pm0.03}$ | $3.27_{\pm0.42}$ | $14.33_{\pm2.84}$ | $0.94_{\pm0.27}$ | $2.28_{\pm0.74}$ | $36.50_{\pm6.23}$ | $0.19_{\pm0.04}$ | $2.25_{\pm0.39}$ | $13.67_{\pm3.85}$ | $0.47_{\pm0.12}$ | $2.02_{\pm0.35}$ |
| DSTPP | $2.86_{\pm1.02}$ | $0.12_{\pm0.04}$ | $2.20_{\pm0.33}$ | $6.38_{\pm1.25}$ | $0.39_{\pm0.16}$ | $1.57_{\pm0.28}$ | $24.93_{\pm4.28}$ | $\mathbf{0.14_{\pm0.05}}$ | $1.16_{\pm0.27}$ | $\mathbf{7.41_{\pm1.36}}$ | $0.36_{\pm0.05}$ | $2.21_{\pm0.45}$ |
| SMASH | $6.13_{\pm0.95}$ | $0.26_{\pm0.04}$ | $2.20_{\pm0.18}$ | $9.26_{\pm2.08}$ | $0.58_{\pm0.06}$ | $1.07_{\pm0.14}$ | $35.96_{\pm5.33}$ | $0.35_{\pm0.07}$ | $2.48_{\pm0.25}$ | $9.25_{\pm1.43}$ | $0.45_{\pm0.12}$ | $1.87_{\pm0.39}$ |
| Ours* | $\mathbf{2.72_{\pm0.62}}$ | $\mathbf{0.07_{\pm0.03}}$ | $\mathbf{0.12_{\pm0.07}}$ | $\mathbf{4.13_{\pm0.75}}$ | $\mathbf{0.24_{\pm0.13}}$ | $\mathbf{0.38_{\pm0.17}}$ | $\mathbf{13.53_{\pm2.88}}$ | $0.15_{\pm0.06}$ | $\mathbf{0.12_{\pm0.00}}$ | $8.11_{\pm1.67}$ | $\mathbf{0.12_{\pm0.03}}$ | $\mathbf{0.14_{\pm0.01}}$ |

*Table 2.* Ablation study on Syn-TPP-1 and Syn-STPP-1. The results on other datasets are reported with analysis in Tab. 4, Appendix E.3.

| UOT. | Bias. | Cost Scale. | Temporal Point Process Datasets | | |
|---|---|---|---|---|---|
| | | | Syn-TPP-1 | | |
| | | | WD ↓ | MMD ↓ | STSD ↓ |
| ✗ | ✗ | ✗ | – | $0.90_{\pm0.01}$ | $0.01_{\pm0.00}$ |
| ✓ | ✗ | ✗ | $66.10_{\pm0.22}$ | $0.92_{\pm0.02}$ | $0.01_{\pm0.00}$ |
| ✓ | ✓ | ✗ | $21.59_{\pm0.72}$ | $0.41_{\pm0.01}$ | $0.02_{\pm0.00}$ |
| ✓ | ✓ | ✓ | $\mathbf{2.68_{\pm0.33}}$ | $\mathbf{0.04_{\pm0.00}}$ | $\mathbf{0.01_{\pm0.00}}$ |
| UOT. | Bias. | Cost Scale. | Spatio-Temporal Point Process Datasets | | |
| | | | Syn-STPP-1 | | |
| | | | WD ↓ | MMD ↓ | STSD ↓ |
| ✗ | ✗ | ✗ | – | $0.31_{\pm0.04}$ | $0.48_{\pm0.05}$ |
| ✓ | ✗ | ✗ | $5.88_{\pm1.25}$ | $0.18_{\pm0.02}$ | $0.33_{\pm0.08}$ |
| ✓ | ✓ | ✗ | $3.91_{\pm0.67}$ | $0.15_{\pm0.02}$ | $0.31_{\pm0.06}$ |
| ✓ | ✓ | ✓ | $\mathbf{2.72_{\pm0.62}}$ | $\mathbf{0.08_{\pm0.03}}$ | $\mathbf{0.12_{\pm0.07}}$ |

Thin and with a broader pattern in diffusion-style point-process modeling: while training can be more computationally demanding, joint non-autoregressive generation may offer advantages in inference efficiency (lower sampling runtimes) and competitive sample quality.

## 6 Conclusion

We have presented a diffusion framework for generating spatio-temporal point processes with variable event counts. By formulating the forward process as a Wasserstein-Fisher-Rao gradient flow of a generalized KL divergence, we obtain a principled noising dynamics that jointly transports and creates/annihilates mass. The reverse process is trained using an unbalanced optimal-transport loss, enabling direct comparison and denoising of measures with differing cardinalities. The model successfully captures complex intensity patterns in synthetic and real-world experiments and provides a theoretically grounded extension of diffusion models to measure-valued data, opening avenues for scalable generative modeling of discrete events in continuous space and time.

## Acknowledgements

This work was supported in part by the Key Program of the National Natural Science Foundation of China (NSFC) under Grant No. 72495131; the Shenzhen Stability Science Program 2023, Shenzhen Key Lab of Multi-Modal Cognitive Computing; the Shenzhen Science and Technology Program No. JCYJ20250604141038013; and the Longgang District Key Laboratory of Intelligent Digital Economy Security.

## Impact Statement

This work advances the modeling and generation of temporal and spatio-temporal event data, which has potential applications in domains such as seismology, urban analytics, epidemiology, and social systems. By enabling robust generation under incomplete observations, the proposed approach may support data analysis and simulation in settings where measurements are sparse or partially missing. At the same time, as with other generative models of human-related event data, there is a risk of misuse for sensitive surveillance or predictive profiling if applied without appropriate safeguards. We emphasize that the framework is intended as a general modeling tool, and responsible deployment should consider domain-specific ethical guidelines, data privacy, and transparency requirements.

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

# Appendix

## A   Algorithm

---
**Algorithm 1** Forward Noising
---

**Input:** Initial particles $N^0$
**for** $k = 1$ **to** $K$ **do**
    *// Drift-diffusion for all atoms $z_i$ in $N^{k-1}$*
    $z_i \leftarrow z_i + \eta_k \nabla_z \log \nu(z_i) + \sqrt{2\eta_k}\varepsilon_i$
    *// Birth-death for all updated atoms $z_i$*
    $B_i \sim \text{Bernoulli}\left(\exp\left(-\gamma\eta_k[-\alpha_k(z_i)]_+\right)\right)$
    $P_i \sim \text{Poisson}\left(\gamma\eta_k[\alpha_k(z_i)]_+\right)$
    $N^k \leftarrow \sum_{i=1}^{N^k} B_i(1 + P_i)\delta_{z_i}$
**end for**
**return** $N^K$

---

---
**Algorithm 2** Reverse Denoising Sampler
---

**Input:** Initial state $\hat{N}^K$
**for** $k = K$ **to** $1$ **do**
    *// Drift-diffusion step for all atoms $z_i$ in $\hat{N}^k$*
    $z_i \leftarrow z_i + \eta_k u_\theta(z_i; N^k, \tau_k) + \sigma_k\sqrt{2\eta_k}\varepsilon_i$
    *// Birth-death step for all updated atoms $z_i$*
    $B_i \sim \text{Bernoulli}\left(\exp(-\gamma\eta_k[-\beta_\theta(z_i; N^k, \tau_k)]_+)\right)$
    $P_i \sim \text{Poisson}\left(\gamma\eta_k[\beta_\theta(z_i; \hat{N}^k, \tau_k)]_+\right)$
    $\hat{N}^{k-1} \leftarrow \sum_{i=1}^{\hat{N}^k} B_i(1 + P_i)\delta_{z_i}$
**end for**
**return** $N_0$

---

---
**Algorithm 3** Training with Entropic Unbalanced OT Loss
---

**Input:** Data distribution $p_{\text{data}}$, total diffusion steps $K$
**while** not converged **do**
    Sample $N^0 \sim p_{\text{data}}$ and $k \sim \text{Unif}\{0, \dots, K-1\}$
    *// Forward noising*
    Run Algorithm 1 for $k+1$ steps to get $N^{k+1}$ (and $N^k$)
    *// Reverse denoising prediction*
    Run Algorithm 2 (single step) to predict $\hat{N}_\theta^k$ from $N^{k+1}$
    *// SGD Update*
    Run SGD step to minimize $L(\theta)$ in (15)
**end while**

---

## B   Derivation of the reverse-time drift–diffusion + reaction form

We sketch the standard algebra leading to the drift–diffusion + reaction representation used in the main text.

**Forward intensity dynamics.** Recall the forward WFR-driven intensity evolution

$$\partial_\tau \lambda_\tau = \Delta\lambda_\tau - \nabla \cdot \left(\lambda_\tau \nabla \log \nu\right) - \lambda_\tau \log \frac{\lambda_\tau}{\nu}. \tag{16}$$

**Formal time reversal.** Formally reversing time yields the terminal-value backward equation, equivalently written as the forward-in-$\tau$ PDE

$$\partial_\tau \lambda_\tau = -\Delta\lambda_\tau + \nabla \cdot \left(\lambda_\tau \nabla \log \nu\right) + \lambda_\tau \log \frac{\lambda_\tau}{\nu}. \tag{17}$$

**Score-based rewriting of the transport term.** Define the intensity score field $s_\tau(z) := \nabla \log \lambda_\tau(z)$. Using $\nabla \log \lambda_\tau = \nabla \lambda_\tau / \lambda_\tau$, we have $\lambda_\tau s_\tau = \nabla \lambda_\tau$ and hence $\nabla \cdot (\lambda_\tau s_\tau) = \Delta \lambda_\tau$. Therefore,

$$-\Delta \lambda_\tau + \nabla \cdot \left( \lambda_\tau \nabla \log \nu \right) = -\nabla \cdot \left( 2\lambda_\tau s_\tau \right) + \Delta \lambda_\tau + \nabla \cdot \left( \lambda_\tau \nabla \log \nu \right) \tag{18}$$

$$= -\nabla \cdot \left( \lambda_\tau \left( 2s_\tau - \nabla \log \nu \right) \right) + \Delta \lambda_\tau. \tag{19}$$

**Drift–diffusion + reaction form.** Substituting (19) into (17) yields

$$\partial_\tau \lambda_\tau = -\nabla \cdot \left( \lambda_\tau \left( 2\nabla \log \lambda_\tau - \nabla \log \nu \right) \right) + \Delta \lambda_\tau + \lambda_\tau \log \frac{\lambda_\tau}{\nu},$$

which is exactly the main-text form (12) with $u_\tau(z) = 2\nabla \log \lambda_\tau(z) - \nabla \log \nu(z)$ and $r_\tau(z) = \log(\lambda_\tau(z)/\nu(z))$.

## C   Forward-process hyperparameters and schedules

**Choice of step sizes $\eta_k$ and scaling $\gamma$.** The diffusion-time grid $\{\tau_k\}$ controls (i) the discretization accuracy of the intensity evolution (9) and (ii) the amount of coordinate noise injected per step through the Langevin update (10). A simple default is a uniform grid $\eta_k = 1/K$, with $K$ chosen large enough that the Euler–Maruyama perturbations and the birth–death updates stay in a small-step regime. More generally, nonuniform grids can be used to match a desired corruption schedule (analogous to $\beta$-schedules in Euclidean diffusion), while retaining the interpretation of $\eta_k$ as a discretization step.

The scalar $\gamma$ calibrates the relative strength of count corruption (birth–death) versus coordinate corruption (MOVE). In practice, $\gamma$ is chosen so that the per-step hazards are moderate:

$$\gamma \eta_k \, b_k(z) \text{ and } \gamma \eta_k \, d_k(z) \quad \text{do not become large on typical } z,$$

which avoids excessive variance in the number of offspring and ensures that the discrete chain tracks the intended drift–diffusion–reaction behavior.

**Choice of the net growth field $\alpha_k$.** In the ideal mean-field dynamics (9), the net growth field is

$$\alpha_\tau(z) = -\log\left( \frac{\lambda^\tau(z)}{\nu(z)} \right),$$

which depends on the (unknown) intermediate intensity $\lambda^\tau$. Accordingly, in discrete time we choose $\alpha_k$ as a tractable approximation with two goals: (i) steer the expected event count toward the reference mass $M_\nu = \int_{\mathcal{Z}} \nu(z) \, dz$, and (ii) keep the birth–death update stable and easy to sample.

A particularly simple instantiation is a *global* net growth rate $\alpha_k(z) \equiv a_k$ chosen to match a desired expected-count schedule $\{\bar{M}_k\}_{k=0}^K$:

$$a_k := \frac{1}{\gamma \eta_k} \log \frac{\bar{M}_k}{\bar{M}_{k-1}}. \tag{20}$$

With this choice, the birth–death step induces an average multiplicative change consistent with $\bar{M}_k / \bar{M}_{k-1}$. We set $\bar{M}_0 := \max(|N^0|, 1)$ and use the geometric interpolation

$$\bar{M}_k := \bar{M}_0^{1-k/K} M_\nu^{k/K}, \qquad k = 0, \dots, K.$$

More expressive choices allow $\alpha_k(z)$ to vary over $\mathcal{Z}$ (e.g., using a coarse estimate of $\lambda^\tau$ relative to $\nu$). In this work we treat $\alpha_k$ and $\gamma$ as forward-process design parameters and study their effect empirically.

**Choice of the reference intensity $\nu$.** We choose $\nu(z)$ as a simple reference intensity on $\mathcal{Z}$ (e.g., spatio-temporally homogeneous). A convenient default is to set $\nu(z) \equiv c$ and choose its total mass $M_\nu$ to match the scale of event counts in the data (e.g., matching the empirical mean count, optionally with $M_\nu$ randomized per sample by a Poisson draw). When $\nu$ is constant, the drift term $\nabla \log \nu(z)$ vanishes and the Langevin move update (10) reduces to isotropic Gaussian perturbations,

$$z_i \leftarrow z_i + \sqrt{2\eta_k}\, \varepsilon_i, \qquad \varepsilon_i \sim \mathcal{N}(0, I),$$

with reflection at the boundary of $\mathcal{Z}$ when the domain is bounded.

*Table 3.* Dataset statistics.

| Category | Dataset | Statistics | | | |
| --- | --- | --- | --- | --- | --- |
| | | # Sequences | Avg. Event Count | Time Horizon | Spatial Scope |
| **Temporal Point Process** | **Syn-TPP-1** | 2000 | 59.22 | 40.00 | – |
| | **Syn-TPP-2** | 2000 | 61.57 | 40.00 | – |
| | **Twitter** | 2019 | 14.90 | 24.00 | – |
| | **Covid-19** | 156 | 2.41 | 7.00 | – |
| **Spatio-Temporal Point Process** | **Syn-STPP-1** | 6000 | 41.25 | 10.00 | $[-1.00, +1.00] \times [-1.00, +1.00]$ |
| | **Syn-STPP-2** | 6000 | 40.08 | 10.00 | $[-1.00, +1.00] \times [-1.00, +1.00]$ |
| | **Crime** | 313 | 134.26 | 168.00 | $[-84.55, -84.28] \times [33.37, 33.89]$ |
| | **Earthquake** | 1500 | 76.27 | 30.00 | $[122.29, 149.86] \times [22.00, 45.94]$ |

# D    Datasets, Baselines, and Computation of Metrics

## D.1    Datasets

We evaluate the proposed model on both temporal point processes (TPPs) and spatio-temporal point processes (STPPs), using synthetic benchmarks with known ground-truth structure and real-world datasets that reflect diverse application domains. Tab. 3 summarizes key statistics.

**Temporal Point Process (TPP)**    For TPPs we consider two synthetic datasets with explicit generative mechanisms and also employ two real-world datasets which exhibit irregular inter-event times and sparse, heterogeneous event counts, offering a realistic testbed for generative modeling of social and policy dynamics.

*(i)* **Syn-TPP-1**: a synthetic temporal point process following a multi-modal inhomogeneous Poisson process. It exhibits pronounced intensity peaks at multiple time locations, allowing us to test the model's ability to recover complex, time-varying intensity patterns. The dataset contains 2000 sequences with an average of 59.22 events per sequence over a fixed horizon of 40 time units.

*(ii)* **Syn-TPP-2**: a synthetic temporal Hawkes process with self-exciting dynamics. Its conditional intensity depends on past events, posing a challenge for capturing temporal dependence and excitation-decay structure. It comprises 2000 sequences, each with about 61.57 events on average over 40 time units.

*(iii)* **Twitter** (Farajtabar et al., 2017; Lüdke et al., 2023): a real-world event stream collected from Twitter, containing timestamps of posts along with textual content. It reflects sparse, irregular social-media activity and is used to evaluate the model on natural, heterogeneous temporal point processes. The dataset includes 2019 sequences with an average of 14.90 events per sequence over 24 hours.

*(iv)* **Covid-19 Policy** (Hale et al., 2021; 2020): a real-world dataset recording the dates of government policy interventions during the COVID-19 pandemic. We define an event as a confirmed-case decline occurring within a 7-day window, with the corresponding date taken as the event time. This yields sequences with low event counts and irregular timing, testing the model's performance on sparse, policy-relevant event streams. The dataset contains 156 sequences, each with an average of 2.41 events over a 7-day horizon.

**Spatio-Temporal Point Process (STPP)**    For STPPs we use two synthetic benchmarks that evolve both in space and time and consider two real-world datasets which exhibit pronounced spatial clustering, variable event counts, and long-range temporal dependencies, challenging the model to capture realistic spatio-temporal structure.

*(i)* **Syn-STPP-1**: a synthetic spatio-temporal point process defined as a multi-modal inhomogeneous Poisson process with evolving spatial modes over time. It is designed to assess joint recovery of time-varying intensity surfaces and the preservation of spatial structure. The dataset consists of 6000 sequences, each with about 41.25 events over 10 time units, within the spatial domain $[1, -1] \times [1, -1]$.

*(ii)* **Syn-STPP-2**: a synthetic spatio-temporal Hawkes process that combines self-excitation in time with spatial triggering. It tests the model's capacity to capture both temporal and spatial dependencies. The dataset contains 6000 sequences, with an average of 40.08 events over 10 time units, also defined on $[1, -1] \times [1, -1]$.

*(iii)* **Crime**: Follow Yang et al., we process a real-world spatio-temporal dataset from the Atlanta Police Department (2015–2020), consisting of geolocated crime incidents. It exhibits strong spatial clustering and irregular temporal patterns, offering a realistic benchmark for urban event data. The dataset includes 313 sequences, each with an average of 134.26 events over 168 hours, within the spatial bounding box of Atlanta.

*(iv)* **Earthquake**: a real-world spatio-temporal dataset from the U.S. Geological Survey, containing locations and times of earthquakes (magnitude $\geq 2.0$) in Japan. It features spatially and temporally clustered events with variable counts, challenging the model to represent long-range dependencies and intense localization. The dataset comprises 1500 sequences, with an average of 76.27 events, across the spatial region covering Japan's seismic activity.

## D.2   Baselines

**Temporal Point Process (TPP)**   We compare the proposed method with representative baselines for temporal point process (TPP) generation, including autoregressive intensity-based models, flow-based models, and diffusion-based approaches. All baselines are trained on the same training splits and evaluated by generating event sequences over the test horizon.

*(i)* **RMTPP** (Du et al., 2016): It is an autoregressive intensity-based model, which models the conditional intensity function using recurrent neural networks and is trained via maximum likelihood estimation. It provides an explicit generative mechanism, and event sequences are generated using standard thinning-based simulation.

*(ii)* **THP** (Zuo et al., 2020): It is an autoregressive intensity-based model, which parameterizes the conditional intensity function with a Transformer architecture to capture long-range temporal dependencies. Following the original work, we generate event sequences by sampling from the learned intensity using thinning or inversion-based methods.

*(iii)* **Add-Thin** (Lüdke et al., 2023): It is a non-autoregressive diffusion-based model that generates event configurations through iterative addition and deletion operations. We follow the original reverse diffusion sampling procedure, using the same number of diffusion steps and noise schedule as reported in the paper, making it the most directly comparable diffusion-based baseline.

*(iv)* **TriTPP** (Shchur et al., 2020): It is a flow-based TPP model, which models event sequences using invertible triangular maps that transform samples from a base Poisson process into the target temporal point process distribution. Event sequences are generated by sampling from the base process and applying the learned triangular transformation, resulting in efficient and exact generation.

For intensity-based and flow-based baselines, event sequences are generated over the test horizon by simulating from the learned conditional intensity or base distribution using standard thinning or transformation-based sampling. For diffusion-based models, samples are obtained via reverse diffusion. To account for sampling variability, all reported results are averaged over multiple independent generations per test sequence.

**Spatio-Temporal Point Process (STPP)**   We compare the proposed method with representative baselines for spatio-temporal point process (STPP) generation, covering likelihood-based models, diffusion-based models, and score-based approaches. All baselines are trained on the same training splits and evaluated by generating event sequences over the test horizon, following the sampling procedures described in their original papers.

*(i)* **DeepSTPP** (Zhou et al., 2022): It is a likelihoos-based STPP model, which parameterizes spatio-temporal event dynamics using neural temporal kernels and spatial interaction functions. Following the original work, we generate event sequences by sampling from the learned conditional intensity using Ogata's thinning algorithm. DeepSTPP has been widely adopted as a generative baseline for spatio-temporal event modeling.

*(ii)* **NSTPP** (Chen et al., 2020): It is a likelihood-based STPP model, which models the conditional intensity function using neural networks and is trained via maximum likelihood estimation. It provides an explicit generative mechanism through standard thinning-based simulation and serves as a canonical likelihood-based STPP baseline.

*(iii)* **DSTPP** (Yuan et al., 2023): It is is a non-autoregressive diffusion-based model that generates event configurations via a learned reverse diffusion process. We follow the original sampling procedure, using the same noise schedule and number of diffusion steps as reported in the paper, making it the most directly comparable baseline to our approach.

*(iv)* **SMASH** (Li et al., 2023): It is a score-matching-based framework originally proposed for pseudolikelihood estimation and uncertainty quantification in marked spatio-temporal point processes. Although SMASH is not explicitly designed

as a generative model, it implicitly defines a conditional intensity function. Following prior practice, we generate event sequences by simulating from the implied intensity using thinning-based sampling.

For all likelihood-based and score-based baselines, event sequences are generated by simulating from the learned conditional intensity over the same test horizon using thinning algorithms. For diffusion-based models, samples are obtained via reverse diffusion. To account for sampling variability, all reported metrics are averaged over multiple independent generations per test sequence.

### D.3 Computation of Metrics

For TPP and STPP generation, we first follow standard practice in point process generation.

*(i)* **WD (Wasserstein distance)**: We first consider **WD (Wasserstein distance)** for both TPP data and STPP data. Define ground truth event count per sequence as $\{K_i\}_{i=1}^n$, and the generated event count per sequence as $\left\{\hat{K}_j\right\}_{j=1}^m$. Then, the WD is defined as

$$W_1 = \frac{1}{n} \sum_{i=1}^{n} \left| K_{(i)} - \hat{K}_{(i)} \right|$$

where $K_{(i)}$ indicates the sorted sample.

*(ii)* **MMD (Maximum Mean Discrepancy over Time and Space)**: As an auxiliary metric, we report the joint Maximum Mean Discrepancy (MMD) to assess the similarity between the real and generated joint spatio-temporal distributions.

Given two dataset $\{(t_i, s_i)\}$ and $\{(\hat{t}_j, \hat{s}_j)\}$, we define a product kernel

$$k\left((t, s), (t', s')\right) = k_t(t, t') \cdot k_s(s, s')$$

where $k_t$ and $k_s$ are radial basis function (RBF) kernels over time and space, respectively.

The joint MMD is computed as

$$\text{MMD}^2 = \mathbb{E}_{x,x'\sim\mu}[k(x, x')] + \mathbb{E}_{y,y'\sim\nu}[k(y, y')] - 2\mathbb{E}_{x\sim\mu,y\sim\nu}[k(x, y)]$$

By using a joint kernel over time and space, this metric captures dependencies between temporal and spatial dimensions, in contrast to marginal comparisons that consider time or space alone.

To evaluate joint spatio-temporal fidelity, we need to capture discrepancies in time, space, and event count in a unified manner.

*(iii)* **STSD (Spatio-Temporal Sinkhorn Divergence)**: Given a real sequence $\mathcal{X} = \{(t_i, s_i)\}_{i=1}^N$, and a generated sequence $\hat{\mathcal{X}} = \left\{(\hat{t}_i, \hat{s}_i)\right\}_{i=1}^{\hat{N}}$, we represent each sequence as a discrete point measure:

$$\mu = \sum_{i=1}^{N} \delta_{(t_i, s_i)}, \quad \nu = \sum_{j=1}^{\hat{N}} \delta_{(\hat{t}_i, \hat{s}_i)}$$

We define the spatio-temporal transport cost as

$$c\left((t, s), (t', s')\right) = \frac{(t - t')^2}{\sigma_t^2} + \frac{\|s - s'\|^2}{\sigma_s^2}$$

where $\delta_t$ and $\delta_s$ control the temporal and spatial length-scales, respectively.

The entropic unbalanced OT cost

$$\mathcal{S}^{\text{ST}}(\mu, \nu) = \text{UOT}_\epsilon(\mu, \nu) = \text{UOT}_\epsilon(\mu, \nu) - \frac{1}{2}\text{UOT}_\epsilon(\mu, \mu) - \frac{1}{2}\text{UOT}_\epsilon(\nu, \nu)$$

This metric compares the joint spatio-temporal geometry of event sequences while remaining robust to differences in sequence length.

# E    Experimental Details

## E.1    Intensity Recovery

Fig. 5 reports results on Syn-TPP-1, a multi-modal inhomogeneous Poisson process. The generated event-time distributions closely match the ground-truth intensity, correctly recovering both the locations and relative magnitudes of multiple modes. Notably, the model captures the global intensity landscape as a whole, rather than approximating it through locally accumulated decisions, highlighting the benefit of configuration-level denoising over autoregressive or local edit-based generation.

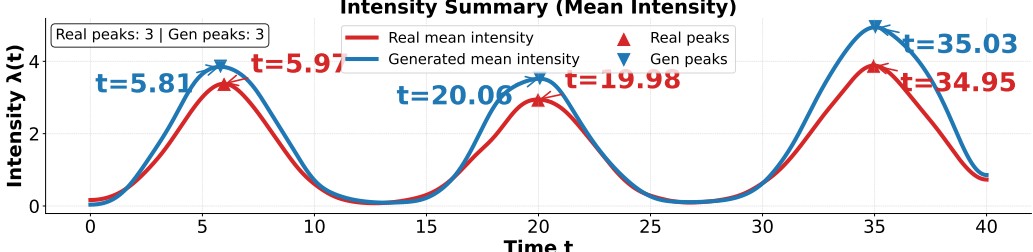

*Figure 5.* Multimodal temporal intensity recovery on Syn-TPP-1.

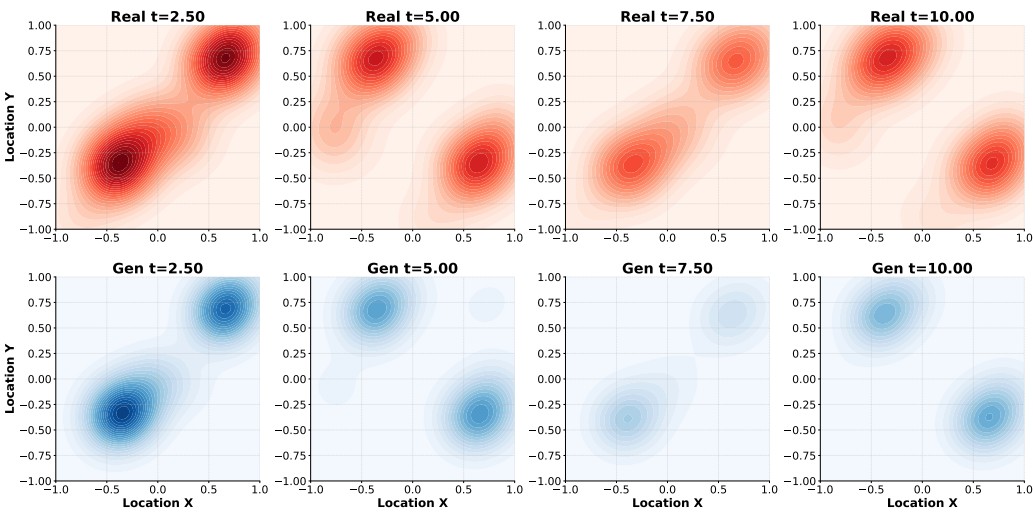

*Figure 6.* Recovery of multimodal spatio-temporal intensity on Syn-STPP-1.

In addition to the intensity recovery shown on Syn-TPP-1, we further evaluate our model on the spatio-temporal dataset Syn-STPP-1. As illustrated in Fig. 6, the learned intensity surfaces at different time steps ($t = 2.5, 5.0, 7.5, 10.0$) successfully capture the underlying multi-modal structure. For example, at $t = 5.0$, two prominent intensity peaks emerge around coordinates $(-0.5, 0.5)$ and $(0.5, -0.5)$, closely matching the ground-truth intensity modes. In short, the recovered intensity not only preserves the spatial multi-modality but also reflects plausible temporal dynamics. These results confirm that our model can faithfully reconstruct complex intensity surfaces, even in challenging spatio-temporal settings where both time and location interact.

## E.2    Conditional Generation

To further evaluate the conditional generation capability of our model, we conduct experiments on a synthetic two-mode inhomogeneous Poisson process designed to simulate diurnal activity patterns under different day-types. In this setup, the intensity function is conditioned on a binary context variable $c$ representing "weekend" and "weekday" conditions, defined as a mixture of two Gaussian peaks with condition-specific location, amplitude, and width parameters. This yields two distinct intensity profiles: **Pattern-1 (weekend)** is characterized by later, broader, and lower peaks, capturing more relaxed

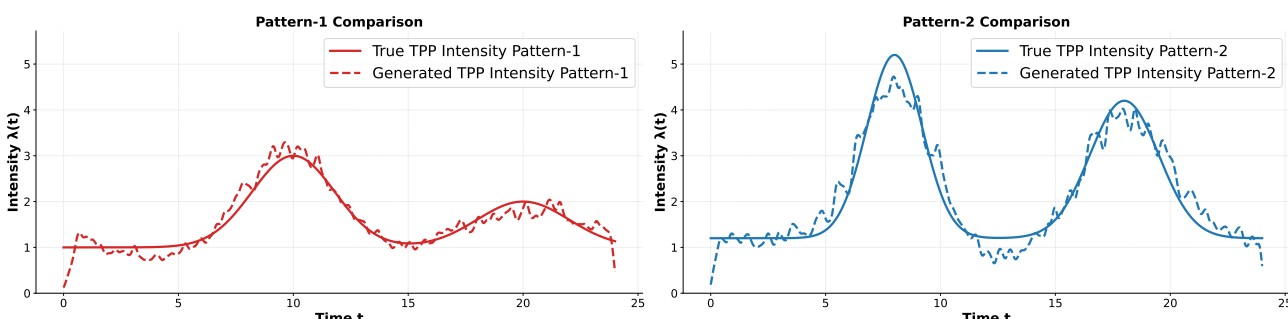

*Figure 7.* Conditional generation on Syn-TPP-1 dataset.

and dispersed activity throughout the day; **Pattern-2 (weekday)** exhibits sharper, higher-magnitude peaks occurring earlier in the day, reflecting concentrated activity periods such as morning and evening commutes.

As illustrated in Fig. 7, our model accurately reconstructs these conditional intensity profiles from the generated sequences. For Pattern-1, the model successfully recovers the later, broader bimodal structure while maintaining the correct relative height and width of the two modes. For Pattern-2, the generated intensity closely matches the ground-truth early, sharp dual peaks, preserving both their timing and amplitude.

These results demonstrate that our diffusion-based point process model not only preserves the overall intensity shape but also faithfully recovers fine-grained temporal structures across distinct conditional patterns. The close agreement in both timing and magnitude underscores the model's ability to learn and reproduce complex, context-dependent temporal dynamics without over-smoothing or mode collapse—a key advantage over prior generative approaches for point processes.

Finally, Fig. 8 evaluates conditional generation on Syn-STPP-1. The generated conditional intensities consistently align with the corresponding ground-truth patterns across different contexts, indicating that the model learns context-dependent generative mechanisms rather than memorizing unconditional statistics. This behavior is enabled by our unified diffusion formulation, which naturally incorporates contextual information at the configuration level and allows global adaptation of event patterns under conditioning.

### E.3 Ablation Study

*Table 4.* Ablation study on key components of the proposed diffusion framework. We ablate the following modules: *(i)* UOT: replace the proposed unbalanced optimal transport (UOT) objective with a balanced OT variant that enforces strict event-count matching between generated and target sample paths, in order to assess the importance of mass flexibility. *(ii)* Bias: ablate the Sinkhorn divergence by removing the debiasing term, e.g., use raw entropic UOT cost $\mathrm{UOT}_{\epsilon,\lambda}(\mu, \nu)$ or the debiased unbalanced Sinkhorn divergence $S_{\epsilon,\lambda}(\mu, \nu)$, evaluating its effect on training bias and generation quality. *(iii)* Cost Scale: we study the role of temporal and spatial cost scaling by removing the scale factors $\delta_t$ and $\delta_s$ study the role of temporal and spatial cost scaling by removing the scale factors. Metrics are reported as Mean ± SD over three runs; best results are in bold and highlighted in purple.

| | | | **Temporal Point Process Datasets** | | | | | | | | | | | |
| --- | --- | --- | --- | --- | --- | --- | --- | --- | --- | --- | --- | --- | --- | --- |
| | | | **Syn-TPP-1** | | | **Syn-TPP-2** | | | **Twitter** | | | **Covid-19** | | |
| UOT. | Bias. | Cost Scale. | WD ↓ | MMD ↓ | STSD ↓ | WD ↓ | MMD ↓ | STSD ↓ | WD ↓ | MMD ↓ | STSD ↓ | WD ↓ | MMD ↓ | STSD ↓ |
| ✗ | ✗ | ✗ | – | $0.90_{\pm0.01}$ | $0.01_{\pm0.00}$ | – | $0.84_{\pm0.01}$ | $0.01_{\pm0.00}$ | – | $0.66_{\pm0.05}$ | $2.92_{\pm0.36}$ | – | $0.94_{\pm0.03}$ | $7.73_{\pm4.09}$ |
| ✓ | ✗ | ✗ | $66.10_{\pm0.22}$ | $0.92_{\pm0.02}$ | $0.01_{\pm0.00}$ | $8.81_{\pm0.04}$ | $0.85_{\pm0.04}$ | $0.01_{\pm0.00}$ | $16.15_{\pm1.04}$ | $0.39_{\pm0.02}$ | $2.75_{\pm0.31}$ | $1.81_{\pm0.05}$ | $0.93_{\pm0.09}$ | $6.09_{\pm3.38}$ |
| ✓ | ✓ | ✗ | $21.59_{\pm0.72}$ | $0.41_{\pm0.01}$ | $0.02_{\pm0.00}$ | $5.10_{\pm0.72}$ | $0.09_{\pm0.01}$ | $0.01_{\pm0.00}$ | $7.46_{\pm1.12}$ | $0.24_{\pm0.01}$ | $0.73_{\pm0.14}$ | $1.20_{\pm0.40}$ | $0.51_{\pm0.04}$ | $4.63_{\pm1.02}$ |
| ✓ | ✓ | ✓ | $\mathbf{2.68_{\pm0.33}}$ | $\mathbf{0.04_{\pm0.00}}$ | $\mathbf{0.01_{\pm0.00}}$ | $\mathbf{3.09_{\pm0.29}}$ | $\mathbf{0.06_{\pm0.00}}$ | $\mathbf{0.01_{\pm0.00}}$ | $\mathbf{6.78_{\pm0.23}}$ | $\mathbf{0.11_{\pm0.01}}$ | $\mathbf{0.24_{\pm0.02}}$ | $\mathbf{0.37_{\pm0.08}}$ | $\mathbf{0.30_{\pm0.13}}$ | $\mathbf{0.21_{\pm0.18}}$ |
| | | | **Spatio-Temporal Point Process Datasets** | | | | | | | | | | | |
| | | | **Syn-STPP-1** | | | **Syn-STPP-2** | | | **Crime** | | | **Earthquake** | | |
| UOT. | Bias. | Cost Scale. | WD ↓ | MMD ↓ | STSD ↓ | WD ↓ | MMD ↓ | STSD ↓ | WD ↓ | MMD ↓ | STSD ↓ | WD ↓ | MMD ↓ | STSD ↓ |
| ✗ | ✗ | ✗ | – | $0.31_{\pm0.04}$ | $0.48_{\pm0.05}$ | – | $0.79_{\pm0.34}$ | $0.82_{\pm0.25}$ | – | $0.42_{\pm0.19}$ | $0.61_{\pm0.23}$ | – | $0.83_{\pm0.15}$ | $0.75_{\pm0.18}$ |
| ✓ | ✗ | ✗ | $5.88_{\pm1.25}$ | $0.18_{\pm0.02}$ | $0.33_{\pm0.05}$ | $8.27_{\pm1.15}$ | $0.63_{\pm0.28}$ | $0.86_{\pm0.21}$ | $19.72_{\pm4.65}$ | $0.41_{\pm0.13}$ | $0.39_{\pm0.17}$ | $14.67_{\pm3.29}$ | $0.68_{\pm0.13}$ | $0.58_{\pm0.11}$ |
| ✓ | ✓ | ✗ | $3.91_{\pm0.67}$ | $0.15_{\pm0.02}$ | $0.31_{\pm0.08}$ | $7.06_{\pm1.23}$ | $0.39_{\pm0.17}$ | $0.52_{\pm0.15}$ | $15.37_{\pm2.78}$ | $0.22_{\pm0.04}$ | $0.27_{\pm0.06}$ | $11.45_{\pm3.06}$ | $0.37_{\pm0.09}$ | $0.42_{\pm0.07}$ |
| ✓ | ✓ | ✓ | $\mathbf{2.72_{\pm0.62}}$ | $\mathbf{0.08_{\pm0.03}}$ | $\mathbf{0.12_{\pm0.07}}$ | $\mathbf{4.13_{\pm0.75}}$ | $\mathbf{0.24_{\pm0.13}}$ | $\mathbf{0.38_{\pm0.17}}$ | $\mathbf{13.53_{\pm2.88}}$ | $\mathbf{0.15_{\pm0.06}}$ | $\mathbf{0.12_{\pm0.00}}$ | $\mathbf{8.11_{\pm1.67}}$ | $\mathbf{0.12_{\pm0.03}}$ | $\mathbf{0.14_{\pm0.01}}$ |

We conduct an ablation study on both TPP and STPP datasets to assess the contribution of key design choices in our framework: (i) unbalanced optimal transport (UOT), (ii) debiased Sinkhorn divergence, and (iii) temporal–spatial cost

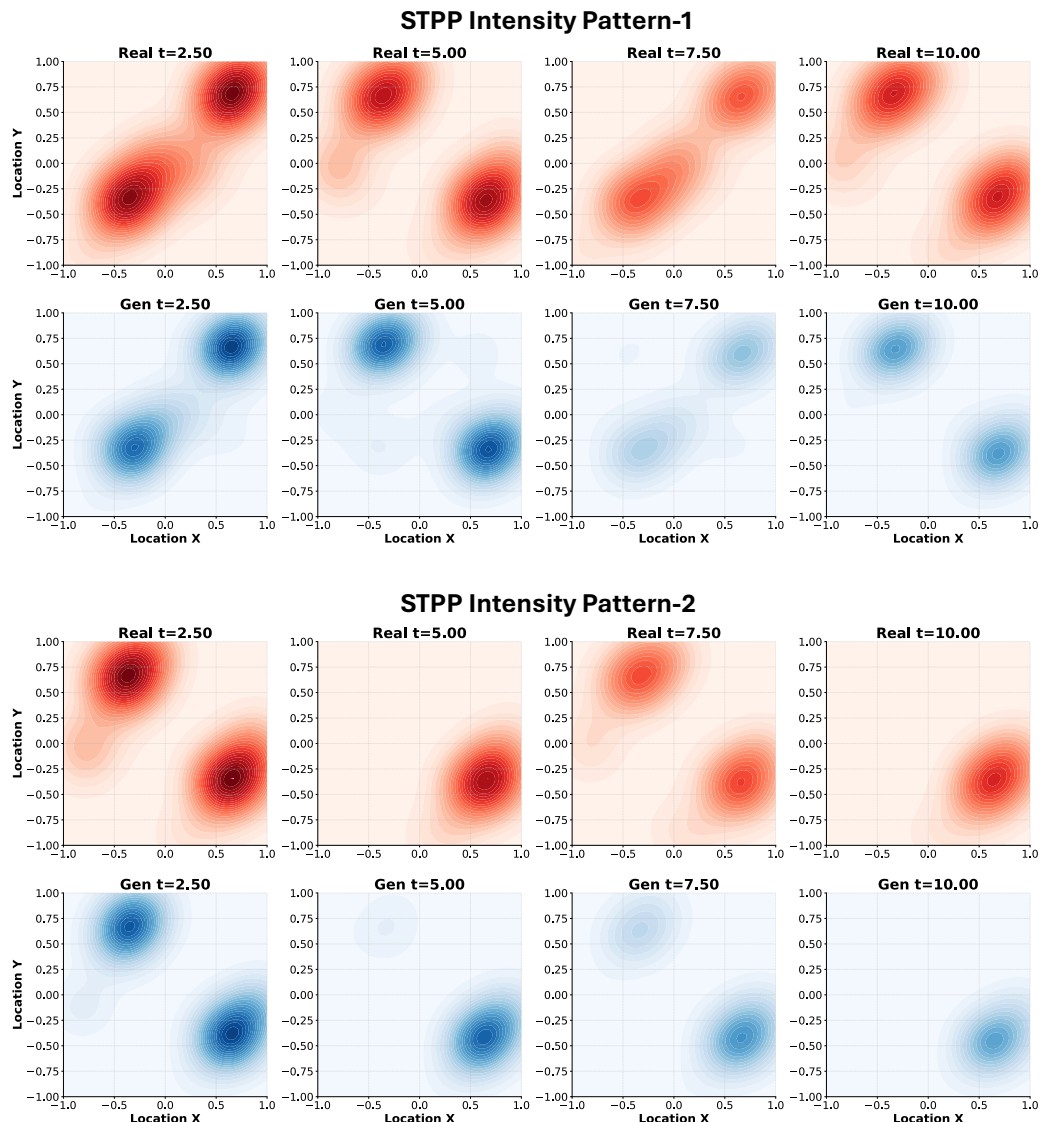

*Figure 8.* Conditional spatio-temporal generation on Syn-STPP-1.

scaling. Tab. 4 summarizes the results.

Replacing the proposed unbalanced optimal transport (UOT) objective with a balanced OT variant that enforces strict event-count matching leads to a substantial degradation in generation quality. This effect varies subtly across different data types. For TPPs, on synthetic datasets with simple intensity patterns, the difference in STSD between UOT and balanced OT remains modest. However, on real-world TPP benchmarks such as Twitter and Covid-19, balanced OT causes a pronounced drop in performance (e.g., STSD increases from 6.09 to 7.73 on Covid-19). This limitation becomes even more acute for spatio-temporal point processes (STPPs). For instance, on Syn-STPP-1, removing UOT raises STSD from 0.33 to 0.48, indicating a clear failure to faithfully reconstruct joint spatio-temporal structure. Similar trends are observed on the Crime and Earthquake datasets. These results collectively underscore the importance of mass flexibility in point process modeling, where event counts are inherently stochastic and should not be rigidly enforced.

Using the raw entropic UOT cost without the debiasing term introduces a systematic bias that negatively affects training stability and sample quality. Across STPP benchmarks, removing the debiasing term consistently worsens all metrics. For example, on Syn-TPP-1, MMD increases from 0.41 to 0.92. On Syn-STPP-2, STSD increases from 0.52 to 0.86, and MMD nearly doubles. This confirms that the debiased Sinkhorn divergence is essential for reliable gradient signals and unbiased sample-path supervision under count mismatch.

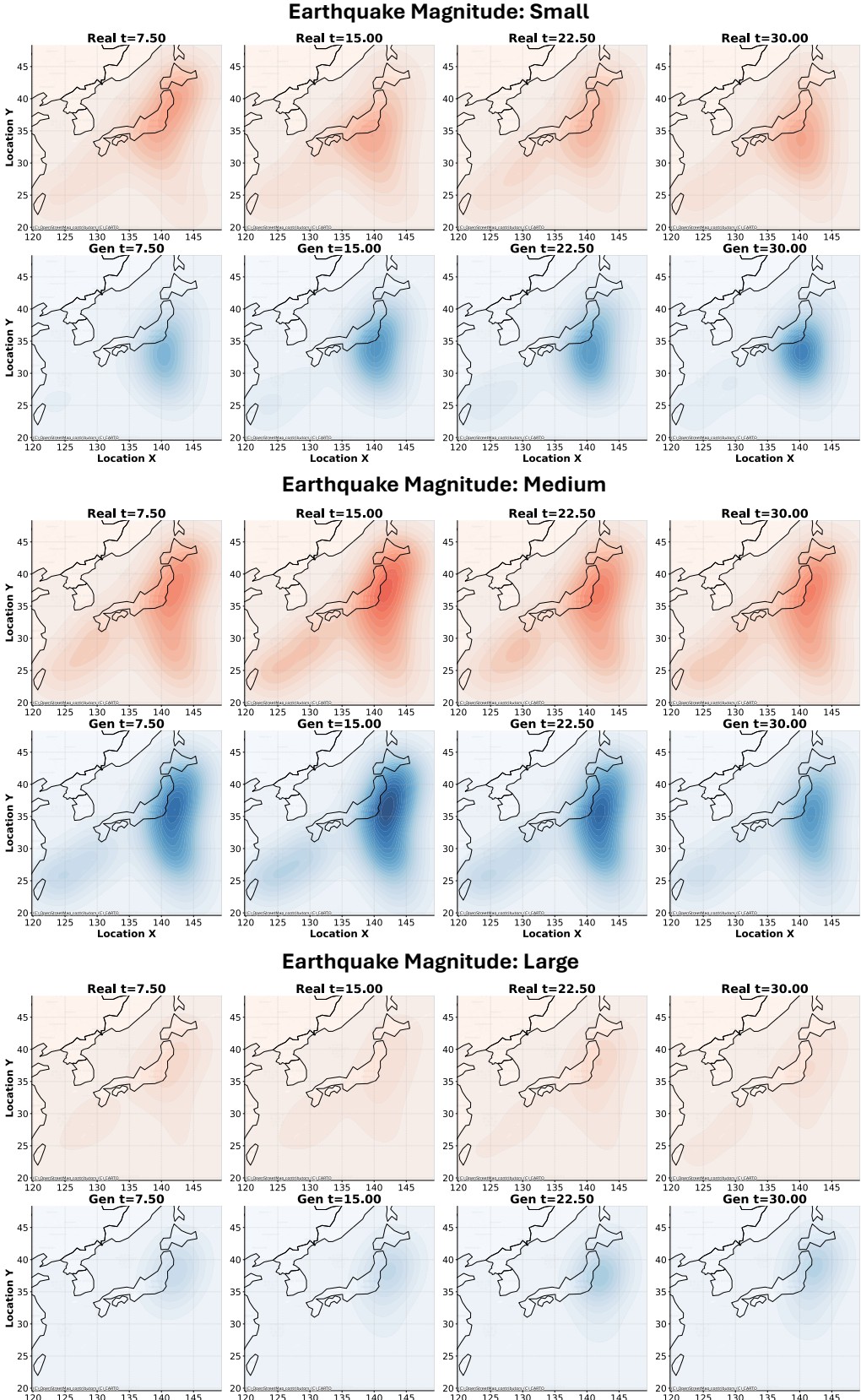

*Figure 9.* Conditional spatio-temporal generation on Earthquake. The conditions are partitioned into three categories based on earthquake magnitude: "small", "medium", and "large".

Removing the temporal and spatial scaling factors leads to a further degradation in performance across both TPP and STPP settings, with the impact being particularly pronounced on real-world datasets. In the temporal setting, disabling the scaling factor causes a marked increase in STSD on the Covid-19 dataset—from 0.21 to 4.63. This substantial decline underscores that properly calibrating the cost scale for time is essential to capturing realistic temporal dynamics. In the spatio-temporal setting, the issue is similarly acute. On the Crime dataset, STSD deteriorates from 0.12 to 0.27 when cost scaling is disabled, indicating that balancing temporal and spatial discrepancies is critical for recovering authentic event dynamics. This effect is especially strong in datasets exhibiting heterogeneous resolutions in time and space.

Taken together, these findings demonstrate that the proposed design choices are not interchangeable implementation details, but are structurally aligned with the geometry of point process configurations, enabling robust learning under variable event counts and complex spatio-temporal dynamics.

### E.4 Scalability

We evaluate the scalability of our model on both Syn-TPP-1 and Syn-STPP-1 by varying the number of training samples (2k–20k) and the event count per sequence (∼10–100). The results are shown in Fig. 10 and Fig. 11. Because our method operates on full event configurations and uses transport-based objectives, it naturally incurs additional computational overhead. The experiments on scalability were intended to illustrate scaling behavior rather than to argue for absolute efficiency. In particular, they show that the per-epoch training time grows moderately with sequence length and dataset size, while remaining in a comparable range relative to other neural TPP/STPP models.

### E.5 Advantages in Inference Time.

As shown in Fig. 4 and discussed in main text, our branching diffusion framework brings inference time advantages. Consistent with the discussion in Add-Thin and with a broader pattern in diffusion-style point-process modeling: while training can be more computationally demanding, joint non-autoregressive generation may offer advantages in inference efficiency and competitive sample quality.

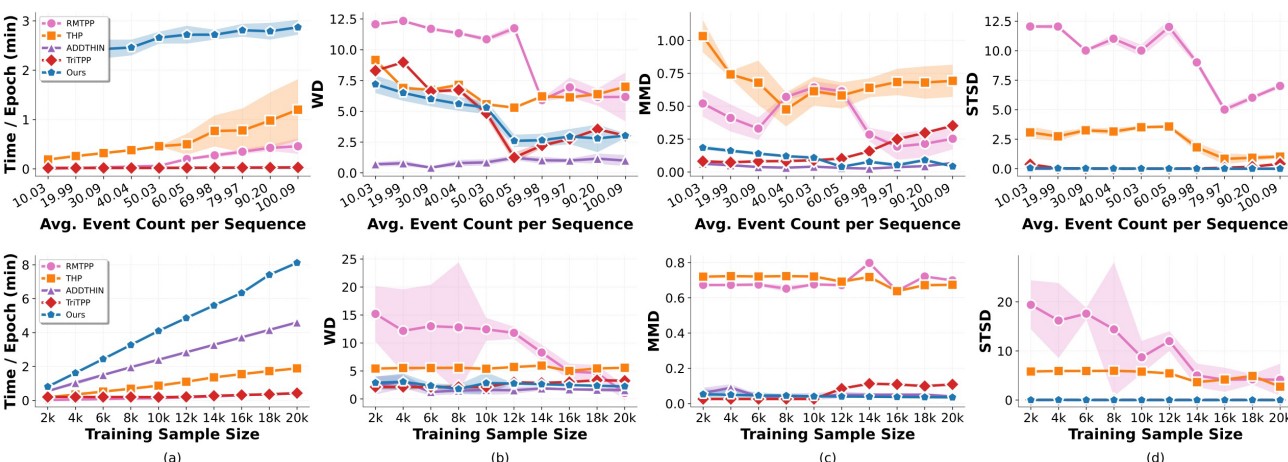

*Figure 10.* Trainig time efficiency and scalability on Syn-TPP-1. All the experiments are conducted over three random runs and the standard error is reflected in the shaded areas. Top row: (a)–(d) Training Time/WD/MMD/STSD vs. average event count per sequence (10.03–100.09. For each case, the training sample size is fixed as 6000). Bottom row: (a)–(d) Training Time/WD/MMD/STSD vs. training sample size (2000–20000, step 2000. For each case, the average event count per sequence is ≈ 60).

## F  Reproducibility Analysis

### F.1  Computing Infrastructure

All synthetic data experiments and real-world data experiments, including the comparison experiments with baselines, were conducted on an Ubuntu 20.04.5 LTS system equipped with NVIDIA GeForce RTX 3090 GPUs. The GPU operates at a maximum clock speed of 2100 MHz for both graphics and streaming multiprocessors (SM), 30 Gigabyte memory.

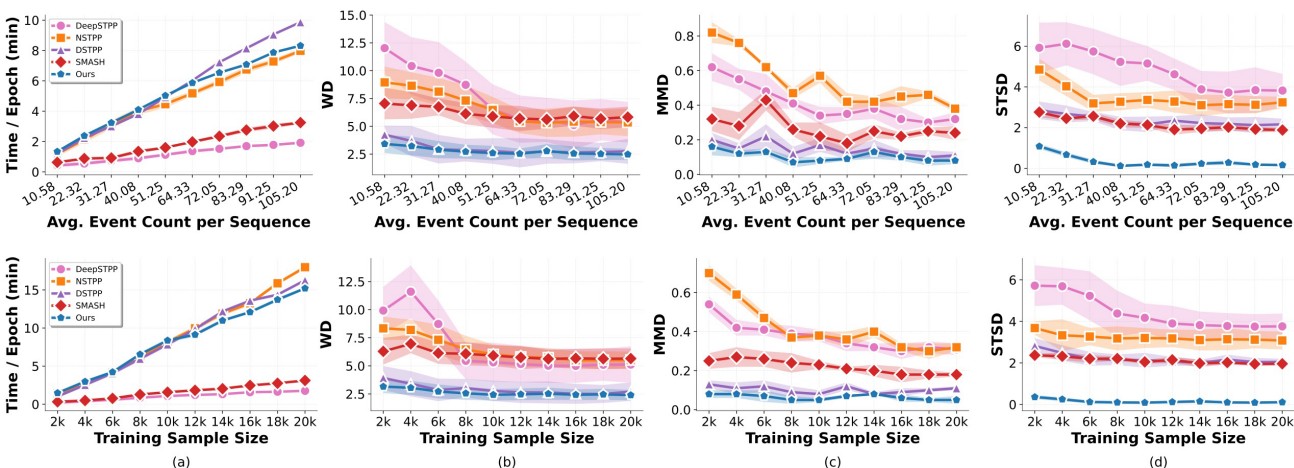

*Figure 11.* Training time efficiency and scalability on Syn-STPP-1. All the experiments are conducted over three random runs and the standard error is reflected in the shaded areas. Top row: (a)–(d) Training Time/WD/MMD/STSD vs. average event count per sequence (10.58–105.20. For each case, the training sample size is fixed as 6000). Bottom row: (a)–(d) Training Time/WD/MMD/STSD vs. training sample size (2000–20000, step 2000. For each case, the average event count per sequence is $\approx 40$).

*Table 5.* Descriptions and values of hyper-parameters used for models trained on the synthetic and real-world datasets.

| Hyper-Parameters | Value Used | | | | | | | |
|---|---|---|---|---|---|---|---|---|
| | **TPP** | | | | **STPP** | | | |
| | **Syn-TPP-1** | **Syn-TPP-2** | **Twitter** | **Covid-19** | **Syn-STPP-1** | **Syn-STPP-2** | **Crime** | **Earthquake** |
| Max Epochs | 50 | 50 | 50 | 50 | 50 | 50 | 50 | 50 |
| Batch Size | 64 | 64 | 64 | 64 | 64 | 64 | 32 | 32 |
| Sinkhorn Iterations ($L$) | 100 | 100 | 100 | 100 | 50 | 50 | 50 | 50 |
| Reverse Steps | 5 | 5 | 5 | 5 | 20 | 20 | 20 | 20 |
| Learning Rate | 1e-3 | 1e-3 | 1e-3 | 1e-3 | 1e-3 | 1e-3 | 1e-3 | 1e-3 |
| Optimizer | Adam | Adam | Adam | Adam | Adam | Adam | Adam | Adam |

### F.2   Hyper-Parameter Selection

The key parameter selection are reported in Tab. 5

## G   Limitation and Broader Impacts

**Limitation**   While the proposed framework provides a principled diffusion formulation for temporal and spatio-temporal point processes, several limitations remain. First, the current implementation relies on entropic unbalanced optimal transport and Sinkhorn iterations during training, which introduce additional computational overhead compared to purely likelihood-based models, especially for large event sets. Second, our reverse-time dynamics are learned under a mean-field approximation of the branching process, which may limit accuracy in settings with extremely strong inter-event dependencies. Finally, although the model generalizes well across the benchmarks considered, extending the framework to very high-dimensional mark spaces or long-horizon sequences may require additional architectural or algorithmic refinements.

**Broader Impacts**   This work advances the modeling and generation of temporal and spatio-temporal event data, which has potential applications in domains such as seismology, urban analytics, epidemiology, and social systems. By enabling robust generation under incomplete observations, the proposed approach may support data analysis and simulation in settings where measurements are sparse or partially missing. At the same time, as with other generative models of human-related event data, there is a risk of misuse for sensitive surveillance or predictive profiling if applied without appropriate safeguards.

We emphasize that the framework is intended as a general modeling tool, and responsible deployment should consider domain-specific ethical guidelines, data privacy, and transparency requirements.

