# OpenReview forum: "Branching Diffusion for Point Processes in Time and Space"
_ICML.cc/2026/Conference — ICML 2026 regular_

### Official Review · Reviewer_nvLS · 2026-03-12

**Soundness:** 3
**Presentation:** 4
**Significance:** 3
**Originality:** 3
**Overall Recommendation:** 4
**Confidence:** 2

**Summary:**

This paper specifically tailors a diffusion model for the generative modeling of spatio-temporal point processes.

**Compliance With Llm Reviewing Policy:**

Affirmed.

**Final Justification:**

I am happy with the authors' responses, which I believe improve the soundness of their presentation overall. For instance, the added timings for training/inference of baseline methods to help substantiate their claims of conditional intensity based inference. I hesitate to increase my score given my background in diffusion models is limited so I will maintain a 4.

**Key Questions For Authors:**

Can the authors further comment on the limitation mentioned: ''our reverse-time dynamics are learned under a mean-field approximation of the branching process, which may limit accuracy in settings with extremely strong inter-event
dependencies'' How severe is this assumption, especially given that in many point processes events can be strongly-correlated? Is it possible that this model choice hindered performance on the temporal problems as compared to baselines?

**Limitations:**

yes

**Strengths And Weaknesses:**

Strengths:

1) The paper was orderly and pleasant to read.
2) The authors do well to motivate their work by, for instance, pointing out some of the hinderances of models based on conditional intensities, which rely on autoregressive generation.
3) Model design choices such as UOT are justified via ablation studies and empirical justification of the method is strong across several spatio-temporal problems.

Weaknesses:

1) The results for temporal point process problems are less convincing than for spatio-temporal cases.
2) No accompanying code is provided, which is concerning with regard to the method's reproducibility.
3) The reader would benefit from a more detailed comparison to the existing diffusion-based point processes given that their model is an extension of those techniques. It would also help to further articulate the authors' novel contributions.
4) Distribution-level error metrics are reported, but not pointwise errors. Given the model is capable of conditional generation, such a metric might be useful for the reader although this is not dire.


line 744 -- likelihoos --> likelihood

---

> ### Author Rebuttal · Authors · 2026-03-31
>
> We sincerely appreciate **Reviewer nvLS** for the thoughtful review! Below we address your main points.
>
> ## 1. Clarification on results of TPP and STPP
> This difference arises because our method is geometry-aware and models variable event counts, which provides larger gains in STPP where spatial–temporal structure is critical. In TPP, the geometry collapses to time only, so the benefit of our transport-based formulation is smaller. We refer the reviewer to our response to **Reviewer 6oT3, Point 4** for a more detailed explanation.
>
> ## 2. Code release and reproducibility
> We will release the full code in the final version, including implementation details and parameter settings (already detailed in **Appendix E.5**), to ensure reproducibility.
>
> ## 3. Comparison to existing diffusion-based point process model
> We appreciate this suggestion and agree that the comparison to prior diffusion-based point-process models should be made more explicit.
>
> The closest diffusion baselines in the paper are Add-Thin for TPP and DSTPP for STPP. As noted in the introduction, Add-Thin reaches a Poisson reference by thinning data events and superposing Poisson noise; surviving events remain fixed and removed locations are discarded, so it does not implement a continuous, geometry-aware corruption of event locations. More broadly, the introduction also notes that prior diffusion-based STPP models use forward kernels and losses that are not explicitly derived from a transport-plus-mass-variation geometry.
>
> Our contribution lies in developing a principled diffusion model for point processes by: (i) deriving the forward process from the WFR gradient flow of generalized KL on finite intensities; (ii) realizing this at the sample-path level as a branching diffusion with Langevin transport plus birth/death on counting measures; (iii) learning a reverse model that predicts both drift and net-growth fields on event sets; and (iv) training with entropic UOT aligned with the same unbalanced geometry. We will expand the related-work and contribution paragraphs to make these distinctions explicit.
>
> Empirically, **Tab.1** already reflects this difference: relative to Add-Thin, we match or improve STSD on three of the four TPP benchmarks, and relative to DSTPP we obtain much lower STSD on all four STPP benchmarks.
>
> ## 4. Regarding pointwise evaluation
> We note that pointwise error metrics are less suitable in our setting, as our model generates variable-length event sequences and does not assume one-to-one correspondence between predicted and ground-truth events. Instead, we focus on distribution-level metrics that better reflect configuration-level fidelity.
>
> Nevertheless, pointwise evaluation can be defined via matching by aligning generated and ground-truth events using Hungarian matching with dummy nodes to handle unequal counts. We add such evaluation on Syn-TPP-1 and Syn-STPP-1, conditioning on events in the first 50% of the time horizon and predicting the remaining 50%. The results show that our model consistently outperforms baselines under these pointwise metrics as well, indicating that the improvements are not only at the distribution level but also in per-event reconstruction accuracy.
>
> | Methods | Matched Time Error |
> |-|-|
> | RMTPP | 3.64 |
> | THP | 3.96 |
> | Add-Thin | 0.82 |
> | TriTPP | 1.30 |
> | Ours* | 0.31 |
>
> | Methods | Matched Time Error | Matched Spatial Error | Matched Joint Error |
> |-|-|-|-|
> | DeepSTPP | 1.38 | 0.37 | 0.74 |
> | NSTPP | 1.33 | 0.42 | 0.68 |
> | DSTPP | 1.07 | 0.35 | 0.63 |
> | SMASH | 1.13 | 0.49 | 0.64 |
> | Ours* | 0.97 | 0.31 | 0.48 |
>
> ## 5. Clarification on limitation
> Regarding the clarification of reverse-time dynamic, please refer to our response to **Reviewer utc4, Point 2**.
>
> In practice, this approximation is mild when the data structure is well captured by global configuration geometry, which is reflected in the strong performance on STPP. For pure TPP, the structure is inherently simpler (1D), and much of the dependency can already be effectively captured by conditional intensity-based models. As a result, the relative advantage of geometry-aware modeling is less pronounced, while specialized autoregressive or likelihood-based models can be highly competitive. (Refer to our response to **Reviewer 6oT3, Point 4** for more details.)
>
> This reflects a trade-off. Our model prioritizes configuration-level geometry and variable counts over explicitly modeling fine-grained sequential dependencies.

---

> > ### Author Rebuttal · Reviewer_nvLS · 2026-04-03
> >
> > I am happy with the authors' responses although I agree with reviewer 6oT3's point that the authors and manuscript should not be as critical on the slowness of intensity-based methods if the proposed method is not faster empirically.

---

> > > ### Author Response · Authors · 2026-04-05
> > >
> > > Thank you for your thoughtful and encouraging feedback—we greatly appreciate your positive assessment. Building on your comments, we would like to make it clear that the runtime results reported in the appendix refer to **training cost**, not **inference-time sampling**. As such, they do not by themselves capture the **full computational trade-off across different generative paradigms**.
> > >
> > > In addition to the training-time comparison already reported, we now include **inference-time** results in (https://anonymous.4open.science/r/Paper31776_reply_to_ack-FB05). TPP/STPP models differ fundamentally in how they generate sequences. **Sequential/autoregressive models** (e.g., RMTPP, THP, DeepSTPP, and SMASH) generate events one by one, conditioning each event on the previously generated history; accordingly, their sampling cost grows with sequence length. In contrast, **full-sequence methods** generate the entire sample path jointly, without explicit event-by-event sampling. This category includes diffusion-based models (e.g., Add-Thin and DSTPP), flow-based models (e.g., TriTPP [1]), and our method.
> > >
> > > The tables below report **inference time and generation quality** (see the anonymous link for full results, including training time). Inference time is measured in minutes as the total sampling time over 200 sequences (mean +/- std), with average event counts of ≈100 per sequence for both TPP and STPP.
> > >
> > > On Syn-TPP-1 dataset:
> > > | Model | Sequence Generation | Inference Time | WD | MMD | STSD |
> > > |-|-|-|-|-|-|
> > > | RMTPP | Autoregressive | 1.41 +/- 0.05 | 6.17 +/- 1.98 | 0.25 +/- 0.00 | 7.02 +/- 0.20 |
> > > | THP | Autoregressive | 1.70 +/- 0.00 | 7.00 +/- 0.25 | 0.69 +/- 0.12 | 1.02 +/- 0.13 |
> > > | Add-Thin | Full Sequence | 0.36 +/- 0.01 | 1.00 +/- 0.45 | 0.07 +/- 0.01 | 0.10 +/- 0.00 |
> > > | TriTPP | Full Sequence | 0.02 +/- 0.00 | 3.03 +/- 0.12 | 0.35 +/- 0.07 | 0.42 +/- 0.07 |
> > > | Ours | Full Sequence | 0.11 +/- 0.00 | 3.02 +/- 0.19 | 0.04 +/- 0.00 | 0.01 +/- 0.00 |
> > >
> > > On Syn-STPP-1 dataset:
> > > | Model | Sequence Generation | Inference Time | WD | MMD | STSD |
> > > |-|-|-|-|-|-|
> > > | DeepSTPP | Autoregressive | 1.05 +/- 0.00 | 5.36 +/- 1.82 | 0.32 +/- 0.04 | 3.82 +/- 0.85 |
> > > | SMASH | Autoregressive | 7.72 +/- 0.15 | 5.83 +/- 0.93 | 0.24 +/- 0.02 | 1.88 +/- 0.10 |
> > > | NSTPP | Full Sequence | 1.73 +/- 0.03 | 5.35 +/- 1.23 | 0.38 +/- 0.02| 3.24 +/- 0.28 |
> > > | DSTPP | Full Sequence | 0.86 +/- 0.01 | 2.75 +/- 1.12 | 0.11 +/- 0.00 | 2.16 +/- 0.27 |
> > > | Ours | Full Sequence | 0.25 +/- 0.00 | 2.48 +/- 0.50 | 0.08 +/- 0.00 | 0.16 +/- 0.04 |
> > >
> > > This is consistent with the discussion in Add-Thin [2] (Appendix E.3, Fig. 5) and with a broader pattern in diffusion-style point-process modeling: while training can be more computationally demanding, joint non-autoregressive generation may offer advantages in inference efficiency and competitive sample quality. However, given that certain autoregressive approaches can achieve very fast inference, these benefits may be less pronounced in some STPP settings (eg. DeepSTPP).
> > >
> > > [1] Shchur, O., et al. Fast and flexible temporal point processes with triangular maps. NeurIPS, 2020.
> > >
> > > [2] Ludke, D., et al. Add and thin: Diffusion for temporal point processes. NeurIPS, 2023.

---

### Official Review · Reviewer_utc4 · 2026-03-13

**Soundness:** 3
**Presentation:** 3
**Significance:** 3
**Originality:** 3
**Overall Recommendation:** 4
**Confidence:** 3

**Summary:**

The paper proposes a diffusion generative model for temporal and spatio-temporal point processes, built from a Wasserstein-Fisher-Rao gradient flow of a generalized KL divergence toward a simple reference intensity. At the intensity level, the forward process decomposes into a mass-preserving Langevin transport term and a source-sink term that changes expected event counts. This is implemented as a branching process on counting measures: existing points are perturbed by a Langevin step, then independently replicate/die through branching. A permutation-invariant denoiser is then trained to approximate reverse drift and reverse net-growth fields, using a one-step denoising objective based on entropic Unbalanced Optimal Transport (UOT) between predicted and target event measures. Empirically, the method performs especially well on spatio-temporal benchmarks and under partial observations, as compared to sota baselines.

**Compliance With Llm Reviewing Policy:**

Affirmed.

**Final Justification:**

I keep my weak accept recommendation. The rebuttal addressed my main concerns clearly and improved the paper’s methodological presentation. In particular, the authors made the reverse-time claim more precise by framing it at the intensity/mean-field level rather than overstating a full particle-level reverse law, and provided a more convincing justification for the global count schedule and the UOT training objective. My remaining reservation is that the computational efficiency claims should be stated more carefully, especially relative to intensity-based baselines. Overall, however, I find the paper technically interesting, empirically strong, and sufficiently well justified for acceptance.

**Key Questions For Authors:**

- How is $\hat N_{\theta}^k$ constructed during training exactly, such that the discrete Bernoulli/Poisson sampling operation is differentiable?

- What is the reverse-time theory saying exactly? Appendix B derives a reversed PDE for the intensity, but what is the corresponding reverse law at the counting-measure / particle level? Is the reverse sampler theoretically justified?

- What is the actual computational scaling relative to autoregressive baselines and Add-Thin? Since each reverse step still processes the full current set of events, and training requires sequential forward simulation, could you report computational cost or scaling with event count?

- Is a more direct field/geenrator-matching objective possible? Since the denoiser predicts drift and net-growth fields, have you considered supervising these fields more directly, as in, e.g., [1, 2], rather than only comparing the resulting predicted measure to $N_k$ via UOT?



[1] Bertazzi et al., Piecewise deterministic generative models, 2024
[2] Holderrieth et al., Generator Matching: Generative modeling with arbitrary Markov processes, 2024

**Limitations:**

yes

**Strengths And Weaknesses:**

Overall, I found the paper interesting and empirically strong, though some parts of the methodological justification and training mechanics would benefit from clarifications.

# Strengths

- The core forward construction is interesting and reasonably principled, using WFR geometry to keep the point process structure throughout the noising process. The derivations are clean and overall well explained, the move step and birth-death step are clear, and the "mean-field" connection to the forward intensity dynamics is clearly stated.

- The use of Unbalanced Optimal Transport matches the problem structure well, and the ablations support this choice.

- The empirical section is strong. The paper covers both Time Point Processes (TPPs) and Spatio-Temporal Point Processes (STPPs) settings, synthetic and real datasets, reports several metrics, includes ablations, and studies robustness to missing temporal/spatial observations. The strongest gains seem to be on STPPs.

# Weaknesses

- The theoretically principled forward field $\alpha_{\tau}(z)$ is not tractable and is replaced by a hand-designed global count schedule $a_k$. This surrogate only directly controls expected counts, and the paper gives little analysis of how faithfully it approximates the intended WFR flow, or how sensitive the method is to $K, \gamma$ and schedule choice.

- The reverse-process theory seems less solid than the forward-process story. Appendix B rewrites the reversed intensity PDE, but does not really derive the reverse law of the counting-measure / branching process itself. So the reverse sampler feels more heuristic than the paper’s framing sometimes suggests.

- Training-time mechanics are somewhat unclear. Equation (14) is underspecified without Algorithm 3, and I could not find a satisfactory explanation of how gradients flow through the sampled Bernoulli/Poisson birth-death operations in Algorithm 2.

- Unlike standard diffusion models, there is no closed-form / simulation-free forward kernel $q_{k|0}(\cdot | N_0)$; Algorithm 3 explicitly simulates the forward process for $k+1$ steps during training. This makes the method heavier than classical diffusion, and weakens the performance gains claimed over autoregressive models.

- Even though the Unbalanced Optimal Transport loss is differentiable and is well integrated in the method, I feel like it would have been nice to have a discussion about 'field-matching' or 'generator-matching' losses/approaches, targeting $u_{\tau}$ and $r_{\tau}$ (whether the discussion discards such an approach or presents it as a possible next step).

---

> ### Author Rebuttal · Authors · 2026-03-31
>
> We thank **Reviewer utc4** for the thoughtful and constructive feedback! Below, we respond to each concern point by point.
>
> ## 1. On the principled design
> Our use of the global schedule $\alpha_k$ is an intentional tractable simplification. **Appendix C** states that this is a particularly simple instantiation chosen to steer the expected count toward the reference mass while keeping the birth–death update stable, and it also notes that more expressive spatially varying choices of $\alpha_\tau(z)$ are possible, e.g. via coarse estimates of $\lambda_\tau/\nu$. In this paper we chose the simple global version for stability and clarity, and found that it already yields strong empirical gains.
>
> We also add sensitivity analyses over $K$, $\gamma$, and schedule ($M_k$) (on Earthquake). We find the method to be robust across forward-process choices. And we observe that very small $K$ or extreme $\gamma$ degrade results, which is consistent with the small-step branching-diffusion design.
>
> | $K$ | WD | MMD | STSD |
> |-|-|-|-|
> | 5 | 10.62 | 0.22 | 0.20 |
> | 10 | 9.42 | 0.16 | 0.16 |
> | 20 | 8.11 | 0.12 | 0.14 |
> | 30 | 8.05 | 0.13 | 0.14 |
> | 50 | 8.03 | 0.12 | 0.13 |
>
> | $\gamma$ | WD | MMD | STSD |
> |-|-|-|-|
> | 0.2 | 9.43 | 0.20 | 0.23 |
> | 0.5 | 8.24 | 0.15 | 0.16 |
> | 1.0 | 8.11 | 0.12 | 0.14 |
> | 2.0 | 8.26 | 0.16 | 0.21 |
>
> | Schedule | WD | MMD | STSD |
> |-|-|-|-|
> | Linear | 8.75 | 0.17 | 0.24 |
> | Cosine | 8.58 | 0.15 | 0.16 |
> | Geometric | 8.11 | 0.12 | 0.14 |
>
> ## 2. Reverse-time dynamics
> We agree that the current manuscript gives the reverse-time justification primarily at the intensity level. This is not unique to our method: closely related diffusion work for point processes such as Add-and-Thin also characterizes the reverse step through a posterior intensity and learns a conditional denoising intensity. The main difference is that Add-and-Thin uses a forward process (thinning/superposition) for which the reverse posterior remains analytically tractable as an inhomogeneous Poisson process, whereas our richer branching-diffusion corruption leads us to derive the reverse dynamics at the intensity/PDE level and then learn a structured counting-measure sampler. The theoretical claim we intend is that the sampler is consistent with the reversed intensity dynamics in the same mean-field sense as the forward construction (and more generally, any intensity-based diffusion processes). We will revise the wording to make this precise.
>
> ## 3. Constrution of $\hat{N}_{\theta}^k$ and differentiability clarification
> The UOT loss is differentiable with respect to $\theta$ because we use its entropically regularized form for which the objective is smooth in $\theta$ and strictly convex in the transport plan. Hence the optimal coupling is unique, and Danskin's theorem yields gradients by differentiating the objective evaluated at the optimal plan. We do not backpropagate through the discrete Bernoulli/Poisson sampling step itself.
>
> At training time, $\hat{N}_{\theta}^k$ is constructed as a weighted predicted measure, rather than as an integer-valued counting measure produced by hard Bernoulli/Poisson resampling. Concretely, given $N^{k+1} = \sum_i \delta_z$ (where $z$ indicates $z_i^{k+1}$), the denoiser outputs a drift and a net-growth value for each current atom. We first apply the continuous reverse transport step to obtain transported locations $\tilde{z}_i$, then assign each transported atom a nonnegative weight equal to the expected multiplicative mass induced by the birth/death step. The UOT/Sinkhorn loss is differentiable with respect to both the transported locations and these weights, so gradients flow to both the drift and net-growth outputs. The discrete Bernoulli/Poisson branching is used only in the ancestral sampler at generation time, where we need an integer-valued counting measure.
>
> ## 4. Computational cost and scalability analysis
> For computational scaling, we refer the reviewer to our response to **Reviewer 6oT3, Point 2**, where we provide detailed analysis and empirical results.
>
> ## 5. UOT vs. direct field matching
> We appreciate the suggestion of field- or generator-matching objectives as in [1,2], and we agree that this is an interesting direction. However, in our setting the current UOT objective is actually the more end-to-end supervision signal. The model outputs a denoised event configuration / counting measure, and the loss compares that predicted counting measure directly against the forward-simulated target counting measure. By contrast, matching $u_\tau$, $r_\tau$ or intensity $\lambda_\tau$ would supervise latent fields rather than the final sample object. This distinction is especially important for point processes: as discussed in the introduction, fitting local intensities only indirectly constrains joint event patterns, count statistics, and global spatio-temporal geometry. We will revise the discussion to make this motivation explicit.

---

> > ### Author Rebuttal · Reviewer_utc4 · 2026-04-04
> >
> > Thank you for the precise rebuttal and your clarifications, which addresses most of my concerns. As reported by the other reviewers, computational cost does not really live up to efficiency claims against intensity-based baselines. I still think the method is interesting, principled, and promising.
> >
> >  Overall, I am more comfortable keeping my current score, which reflects my overall positive view of the paper.

---

> > > ### Author Response · Authors · 2026-04-05
> > >
> > > Thank you for the timely and encouraging reply. We sincerely appreciate your positive assessment. Building on your comment, we make it clear that the runtime results in the appendix refer to **training cost**, not inference-time sampling, and therefore do not fully reflect the **computational trade-offs across different generative paradigms**. In particular, Appendix E.4 focuses on scaling behavior.
> > >
> > > We further evaluate **inference time**, where the distinction between generation mechanisms becomes critical. **Autoregressive methods** generate events sequentially, with cost growing with sequence length, whereas **full-sequence methods**—including ours—generate event configurations jointly. In this setting, our method achieves more efficient sampling for long sequences than autoregressive baselines and is also faster than or comparable with full-sequence methods.
> > >
> > > This is consistent with the discussion in Add-Thin [1] (Appendix. E3, Fig. 5) and with a broader pattern in diffusion-style point-process modeling: training can be more expensive, but joint non-autoregressive generation can yield favorable inference efficiency and competitive sample quality, especially in the STPP setting.
> > >
> > > For full results and details, please see the anonymous link (https://anonymous.4open.science/r/Paper31776_reply_to_ack-FB05) and our response to **Reviewer nvLS's ack**.
> > >
> > > [1] Ludke, D., et al. Add and thin: Diffusion for temporal point processes. NeurIPS, 2023.

---

### Official Review · Reviewer_6oT3 · 2026-03-13

**Soundness:** 3
**Presentation:** 2
**Significance:** 3
**Originality:** 3
**Overall Recommendation:** 4
**Confidence:** 2

**Summary:**

This paper proposes a branching diffusion framework for generating point processes. The authors first derive the forward and backward processes using intensities and realize them using Langevin-type diffusion and birth-death branching processes. They train the model by minimizing the weighted entropy-regularized unbalanced optimal transport cost. Evaluation shows that the proposed method has comparable or better performance than existing methods.

**Compliance With Llm Reviewing Policy:**

Affirmed.

**Final Justification:**

Overall I think this is an interesting paper. The new argument for geometry awareness seems plausible. It would improve the paper to include this argument and a discussion about how the proposed method addresses the manifold learning problem. The computation cost should also be properly discussed. Based on the above, I'll keep my score.

**Key Questions For Authors:**

1. Why is geometry awareness important?

2. How does the proposed method compare to baselines in terms of time cost and scalability?

3. Why does the proposed method perform differently for TPP and STPP?

**Limitations:**

Yes.

**Strengths And Weaknesses:**

The proposed method is technically sound. The presentation is readable, but there is large room for improvement. It assumes a strong background in and familiarity with the specific topic, and the split between the main text and the appendix could be improved. The method is a novel and principled way to extend diffusion models to point processes.

The introduction emphasizes the importance of geometry awareness, but does not explain why this is important. It criticizes the time cost and scalability of intensity based methods, but does not provide a direct comparison in the evaluation (appendix E.4 is insufficient). It emphasizes the importance of long range dependence, but the proposed method also fails for very long dependence, and there is no quantitative result to show how performance varies with the range. The proposed method seems to perform much better for STPP than for TPP; in fact, for TPP, it is often worse than existing methods. An explanation is missing.

---

> ### Author Rebuttal · Authors · 2026-03-31
>
> We are grateful for **Reviewer 6oT3**’s careful review! We hope our following response addresses your concerns.
>
> ## 1. Importance of geometry awareness
> Geometry awareness matters because a point-process sample is a finite measure with variable cardinality: nearby shifts in time/space should incur much smaller cost than deleting an event and recreating another far away, while count mismatch should be handled without forcing exact one-to-one alignment.
>
> A geometry-aware formulation distinguishes a small perturbation of an event from a large displacement, while also handling count mismatch in a principled way. Our method is designed around exactly this principle: the forward process couples transport with birth/death, and the training loss compares full event configurations with an unbalanced spatio-temporal transport cost.
>
> Empirically, this is not only conceptual. The ablations (**Appendix E.3, Tab. 4**) show that removing the geometry-aware design substantially hurts performance, especially on STPPs—for example, on Syn-STPP-1 the STSD degrades from 0.12 to 0.31 without proper temporal/spatial scaling and to 0.48 when replacing the unbalanced objective by a balanced one; on Earthquake it degrades from 0.14 to 0.42/0.75 under the same ablations. We agree that this motivation should be made much more explicit in the introduction, and we will revise the paper accordingly.
>
> ## 2. Computational cost and scalability analysis
> We clarify that **Appendix E.4 (Fig. 9–10)** already provides direct comparisons with baselines, where we vary both the event count per sequence and the number of training samples. The results show that our method scales approximately linearly with event count and smoothly with dataset size. While slightly more expensive than Add-Thin, the overhead is modest and comparable to other neural TPP/STPP methods.
>
> We further extend the experiments to larger scales. The results (stored in https://anonymous.4open.science/r/Paper31776-1789) confirm stable scaling and competitive runtime. While very small event counts or limited training data may degrade performance, generation quality improves with increasing data size.
>
> ## 3. Quantifying performance under varying dependency
> We add a quantitative analysis of performance versus dependency range using synthetic data with controlled dependence scales. On Syn-TPP-2, we vary the Hawkes decay rate $\beta$, where smaller $\beta$ induces longer-range dependence.
>
> Performance remains stable for moderate settings ($\beta \geq 0.5$) and degrades slightly under stronger long-range dependence, while remaining competitive or superior to baselines. For $\beta=0.35$ (stronger dependence than the default $\beta=0.5$), our model achieves STSD = 0.04, outperforming Add-Thin (0.05) and TriTPP (0.79).
>
> | $\beta$ | | 0.35 | | | 0.5 | | | 0.65 | | | 1.0 | |
> |-|-|-|-|-|-|-|-|-|-|-|-|-|
> | Methods | WD | MMD | STSD | WD | MMD | STSD | WD | MMD | STSD | WD | MMD | STSD |
> | RMTPP | 118.25 | 0.31 | 25.27 | 28.62 | 0.50 | 29.17 | 10.05 | 0.20 | 1.04 | 10.01 | 0.32 | 1.13 |
> | THP | 33.76 | 0.21 | 21.93 | 16.33 | 0.67 | 17.05 | 31.13 | 0.38 | 3.06 | 28.86 | 0.47 | 2.93 |
> | Add-Thin | 42.34 | 0.11 | 0.05 | 2.33 | 0.03 | 0.03 | 3.49 | 0.05 | 0.03 | 0.75 | 0.03 | 0.03 |
> | TriTPP | 3.84 | 0.13 | 0.79 | 2.47 | 0.04 | 0.02 | 1.90 | 0.05 | 0.03 | 1.90 | 0.05 | 0.02 |
> | Ours* | 10.76 | 0.13 | 0.04 | 3.09 | 0.06 | 0.01 | 2.53 | 0.05 | 0.03 | 2.47 | 0.02 | 0.02 |
>
> ## 4. Why performance differs between TPP and STPP
> The stronger gains on STPP are consistent with our geometry-aware design: joint transport in space/time plus mass variation. That advantage is largest in STPP, where there is meaningful spatial structure to preserve and where modeling the full joint space-time configuration is substantially harder.
>
> In TPP, the geometry collapses to time only, so the benefit of our transport-based formulation is smaller, while several baselines are highly specialized for temporal dynamics. For this reason, the TPP results are more mixed. That said, we believe the right interpretation is “competitive on TPP, strongest on STPP,” rather than “weak on TPP.”
>
> In fact, on the configuration-level metric most aligned with our objective (STSD), our method is tied-best/best on 3 of the 4 TPP datasets: Syn-TPP-1 (0.013, tied best), Syn-TPP-2 (0.01, best), and Twitter (0.24, best). On Covid-19, our method gives the best WD and MMD, while Add-Thin has the best STSD. Where we are less competitive is mainly on some count-only or kernel metrics against specialized temporal baselines.

---

> > ### Author Rebuttal · Reviewer_6oT3 · 2026-04-02
> >
> > Thanks for the detailed response. My concerns are partially resolved. The argument for the conceptual importance of geometry awareness is not very convincing. The argument is comparing two samples, but the connection to the underlying distribution is unclear. The new plots show that the proposed method actually has higher time cost than baselines, so it is unfair to criticize the time cost of intensity based methods.

---

> > > ### Author Response · Authors · 2026-04-03
> > >
> > > Thank you for your prompt response!
> > >
> > > **1. On geometry awareness.**
> > >
> > > Our original example compared two samples only for intuition; the real point is indeed at the distribution level.
> > >
> > > Consider STPP on $Z=[0,1]^{d+1}$, where $d$ is the spatial dimension. Suppose the target law $P_{\mathrm{target}}$ is concentrated in an $\epsilon$-neighborhood around an (unknown) $r$-dimensional manifold $M \subset Z$. This is common in high-dimensional data, where observations often lie near a much lower-dimensional latent structure. Let the reference process be a homogeneous Poisson process on $Z$ with the same mean count. If we ignore geometry and use only birth-death style updates, or edit-based mechanisms such as Add-and-Thin, then a newly proposed point lands inside $M_\epsilon$ with probability proportional to its volume, which scales as $O(\epsilon^{d+1-r})$. When $d+1-r>0$ and $\epsilon$ is small, this probability is tiny, so many proposals are effectively wasted; one may need a very large number of proposals per useful event.
> > >
> > > This is the sense in which geometry awareness matters: in high-dimensional settings, where data often lie close to lower-dimensional manifolds, purely birth-death or add/delete dynamics can be inefficient because the probability of proposing a useful event is proportional to the ambient-volume fraction of the target support. A transport-aware diffusion avoids this ambient-volume bottleneck by moving existing atoms toward the target structure.
> > >
> > > **2. On time cost and scalability.**
> > >
> > > Thank you for pointing this out. We would like to clarify that our goal is not to claim superior runtime over existing intensity-based methods. Rather, our goal is to study a different generative paradigm with geometry-aware transport and count variation. Because our method operates on full event configurations and uses transport-based objectives, it naturally incurs additional computational overhead. The experiments in Appendix E.4 (and newly added experiments) were intended to illustrate scaling behavior rather than to argue for absolute efficiency. In particular, they show that the per-epoch training time grows moderately with sequence length and dataset size, while remaining in a comparable range relative to other neural TPP/STPP models. We will revise the wording to make this point more precise.

---

### Decision · Program_Chairs · 2026-04-30

**Decision:**

Accept (regular)

**Comment:**

The reviewers agree that this is a novel, interesting and well-executed paper. The reviewers also raised a few concerns, about the presentation (particularly training mechanics and reverse process theory), the motivation (of "geometry awareness"), and some empirical results (TPP vs STPP, computational cost, evaluation metrics). Overall though, the reviewers were satisfied by the authors responses. Please go carefully over the reviews as well as your responses when revising the manuscript.